# Emerging signals of declining forest resilience under climate change

Giovanni Forzieri[1,5 ✉], Vasilis Dakos[2], Nate G. McDowell[3,4], Alkama Ramdane[1] & Alessandro Cescatti[1]

Forest ecosystems depend on their capacity to withstand and recover from natural and anthropogenic perturbations (that is, their resilience)[1]. Experimental evidence of sudden increases in tree mortality is raising concerns about variation in forest resilience[2], yet little is known about how it is evolving in response to climate change. Here we integrate satellite-based vegetation indices with machine learning to show how forest resilience, quantified in terms of critical slowing down indicators[3–5], has changed during the period 2000–2020. We show that tropical, arid and temperate forests are experiencing a significant decline in resilience, probably related to increased water limitations and climate variability. By contrast, boreal forests show divergent local patterns with an average increasing trend in resilience, probably benefiting from warming and $CO_2$ fertilization, which may outweigh the adverse effects of climate change. These patterns emerge consistently in both managed and intact forests, corroborating the existence of common large-scale climate drivers. Reductions in resilience are statistically linked to abrupt declines in forest primary productivity, occurring in response to slow drifting towards a critical resilience threshold. Approximately 23% of intact undisturbed forests, corresponding to 3.32 Pg C of gross primary productivity, have already reached a critical threshold and are experiencing a further degradation in resilience. Together, these signals reveal a widespread decline in the capacity of forests to withstand perturbation that should be accounted for in the design of land-based mitigation and adaptation plans.

Forests cover about 41 million km[2] – about 30% of the land surface. They play a fundamental role in the global carbon cycle, absorbing about 33% of anthropogenic carbon emissions, and are considered a key element for mitigating future climate change[6]. In addition, forests provide a series of ecosystem services that contribute to societal well-being, such as regulation of water flows, protection of soils and conservation of biodiversity[7]. Unfortunately, forest ecosystems are increasingly endangered by numerous disturbances, including natural agents (for example, fires, wind storms and pathogens) and anthropogenic pressures[2]. The persistence and functionality of these ecosystems are highly dependent on their resilience, defined as the ability to withstand and recover from environmental perturbations[3–5]. Low-resilience forests are more sensitive to anomalies in external drivers and are potentially more exposed to abrupt and possibly irreversible shifts (for example, regime shifts)[8]. This is particularly critical in view of the ongoing intensification of disturbance regimes that could affect the provision of key ecosystem services in the near future[9–11]. At the same time, forest-based mitigation strategies that rely on sustained carbon sinks and stocks are becoming crucial to achieve the most ambitious climate targets. In this context, it is increasingly important to investigate the vulnerability of forest carbon stocks and fluxes to external perturbations. However, little is known about how forest resilience has been evolving in response to global environmental change. Understanding the underlying mechanisms of forest resilience and its recent dynamics is therefore of paramount importance to develop sound conservation and management plans.

Theoretical studies have demonstrated that as systems approach a tipping point (that is, a threshold when a self-sustained runaway change starts), they lose resilience, so that small continuous external perturbations can shift the system into an alternative configuration[12]. It has been proposed that such a loss of resilience can be detected from the increased temporal autocorrelation (TAC) in the state of the system, reflecting a decline in recovery rates due to the critical slowing down (CSD) of system processes that occur at thresholds[3–5] (Supplementary Methods 1–3 and Supplementary Figs. 1 and 2). In such a framework, resilience is defined as the capacity of ecosystems to withstand perturbations and avoid state shifts, and not as the recovery to the initial state after a state change is induced by a major event. The reduction in resilience can be caused by impaired physiological functions that make the ecosystem unstable or at least more vulnerable to regime shifts under perturbations (for example, in terms of productivity, leaf area index or species composition)[12–14]. This property was leveraged in previous studies to assess spatial patterns of static forest resilience[15–18]. However, application of this method at large scales in a dynamic context

[1]European Commission, Joint Research Centre, Ispra, Italy. [2]Institut des Sciences de l'Evolution de Montpellier (ISEM), Université de Montpellier, CNRS, IRD, EPHE, Montpellier, France. [3]Atmospheric Sciences and Global Change Division, Pacific Northwest National Laboratory, Richland, WA, USA. [4]School of Biological Sciences, Washington State University, Pullman, WA, USA. [5]Present address: Department of Civil and Environmental Engineering, University of Florence, Florence, Italy. ✉e-mail: giovanni.forzieri@unifi.it

is challenging owing to the limited time series of observations, the presence of dominant seasonal frequencies in variations of both ecosystem responses and forcing signals, variations in autocorrelation of the forcing signals and the presence of stochastic noise[4]. So far these challenges have limited the study of the temporal evolution of forest resilience in real systems[19–21] and led to the substantial lack of global-scale assessments. In this respect, the expanding availability of temporally consistent Earth observations over multiple decades is now offering new opportunities to monitor time-varying forest resilience at regional to global scales.

Here we estimate CSD from time series of satellite-based vegetation indices to investigate the space–time variation in forest resilience that has occurred over the past two decades at the global scale. Specifically, we retrieved the 1-lag TAC as a CSD indicator related to resilience[3–5] from satellite-based retrievals of the kernel normalized difference vegetation index (kNDVI) derived for the 2000–2020 period at the global scale at 0.05° spatial resolution from the Moderate Resolution Imaging Spectroradiometer sensor. kNDVI has recently been proposed as a robust proxy for ecosystem productivity[22], and is therefore used in this study as a suitable metric to represent the state of forest ecosystems.

## Trends and drivers of forest resilience

We initially explored the average TAC at the pixel level from the whole kNDVI time series (2000–2020; hereafter referred to as long-term TAC). This signal, by integrating the interplay between forest and climate, reflects the slowness of the forest–climate system resulting from the interplay of environmental drivers that affect plant growth and of the ecosystem capacity to recover from perturbations. A random forest (RF) regression model[23] was then developed to identify the emergent relationships between long-term TAC (response variable) and a suite of forest and climate metrics (environmental predictors; Methods and Extended Data Table 1). Results show that global forests are characterized by a considerable spatial variability in long-term TAC (Extended Data Fig. 1) largely explained by local environmental conditions ($R^2 = 0.87$; Extended Data Fig. 2 and Supplementary Discussion 1). To detect the resilience signal of the forest system and explore its temporal dynamics in response to changing environmental conditions, we analysed the temporal evolution of TAC computed on kNDVI with 3-year rolling windows over the observational period. Factorial simulations of the previously developed RF model were performed to disentangle the contribution of the environmental factors and filter out the confounding signals originating from the TAC of climate drivers (details in Methods). This resulted in a time series of annual TAC and its temporal trend ($\delta$TAC) was used as an indicator of CSD to detect changes in forest resilience over time.

Results show a widespread and significant increase in TAC, and thus a temporal decline in resilience, in tropical, temperate and arid regions ($1.63 \times 10^{-3}$, $1.43 \times 10^{-3}$ and $1.26 \times 10^{-3}$ yr$^{-1}$, respectively). By contrast, boreal forests show divergent local patterns with an average increasing trend in resilience ($-1.54 \times 10^{-3}$ yr$^{-1}$; Fig. 1a,b and Extended Data Table 2) prominently associated with a decline in TAC occurring in Eastern Canada and European Russia. We further explored the temporal changes in resilience, by comparing the average TAC of kNDVI computed over two independent temporal windows (2000–2010 and 2011–2020; Methods). We found a statistically significant increase over time at the global scale (53% of the globe experiences a positive relative change; Fig. 2c). However, the global signal is limited by the compensation of contrasting patterns across different climate regions. In fact, the statistically significant increase of TAC in tropical, arid and temperate forests (56–63% of land with positive relative change) is partially offset by an opposite trend occurring in boreal forests (56% of land with negative relative change). The patterns deriving from the comparison of the two independent decades are consistent with the trajectories of $\delta$TAC (Fig. 1a,b and Extended Data Fig. 3), confirming

the validity of the finding. These emerging signals suggest worrying trajectories for the resilience of much of global forests. The signals are particularly robust because they are based on a single sensor (the Moderate Resolution Imaging Spectroradiometer) and a vegetation index (kNDVI) that showed enhanced correlation with primary productivity and reduced noise and stability issues compared to other classical indices[22] (Methods). Extensive sensitivity analyses further support the robustness of these emerging temporal drifts (Methods, Supplementary Discussion 2 and Extended Data Figs. 4–6).

Looking at the marginal contribution of the drivers of $\delta$TAC, we found that the widespread vegetation greening that occurred in recent decades (Extended Data Fig. 2c and Extended Data Fig. 7a), probably driven by $CO_2$ fertilization and climate change[24], had a positive effect on global resilience, most prominently in cold and temperate climates (Fig. 1d,e, forest density). However, the concurrent intensifications of water limitations and extreme climate events, particularly severe in tropical, arid and temperate regions (Extended Data Fig. 2d,e and Extended Data Fig. 7c–j) have offset the benefits of $CO_2$ fertilization and greening (Fig. 1d,e; |$\delta$TAC| due to changes in background climate and climate variability > |$\delta$TAC| due to changes in forest density). This ultimately resulted in a net loss in forest resilience in these biomes (Fig. 1a–c). The increasing forest vulnerability to natural disturbances and the increased tree mortality throughout much of the Americas and in Europe over recent decades provide independent evidence of ongoing decline of forest resilience[25,26]. The above-mentioned climate-related pressures have occurred in boreal forests as well, but their severity probably could not compensate the gain associated with the positive effect of $CO_2$ fertilization and a warmer climate in most areas of this temperature-limited biome (Fig. 1d,e). However, the pattern observed at the high latitudes could eventually change in response to the expected decline in water availability due to the interplay between global warming and anticipated phenology[27]. In fact, recent observational studies suggest that global forests are switching from a period dominated by the positive effects of $CO_2$ fertilization to a period characterized by the progressive saturation of the positive effects of fertilization on carbon sinks and the rise of negative impacts of climate change[28,29].

## Forest management and resilience

The results shown thus far have focused on the role of natural drivers in modulating spatial and temporal variations in forest resilience. However, anthropogenic disturbances, such as forest management and land use change, have the potential to influence the ability of forest ecosystems to recover from perturbations by directly affecting tree species, age distribution, cover density, rooting depth and primary productivity[1,30,31] (Extended Data Fig. 2c and Extended Data Fig. 7a,b). To factor out such effects, we analysed long-term TAC and $\delta$TAC for managed and intact forests under similar background climate (Methods and Extended Data Fig. 8). Intact forests have considerably lower long-term TAC (that is, higher forest resilience) than managed forests (0.13 and 0.21, respectively; Fig. 2a). This finding reinforces the expectation that intact forests have a higher capacity to withstand external perturbations thanks to their typically higher structural complexity and species richness[32,33]. Independent observational evidence emphasizes the contribution of human pressures in the decline of forest resilience over recent decades[1,26,30,34]. Interestingly, in terms of temporal trends ($\delta$TAC), managed and intact forests do not present significant differences and show comparable fractions of forests experiencing positive trends (72% and 66%, respectively, Fig. 2b) and hence decreasing resilience. This is an important finding because it suggests that the average level of forest resilience in a given climate is heavily affected by forest management, whereas its ongoing temporal variations (Fig. 1a,b) are controlled by large-scale climate signals. The observed global trends, therefore, plausibly reflect the effective climate-induced changes in the capacity of forests to withstand external perturbations.

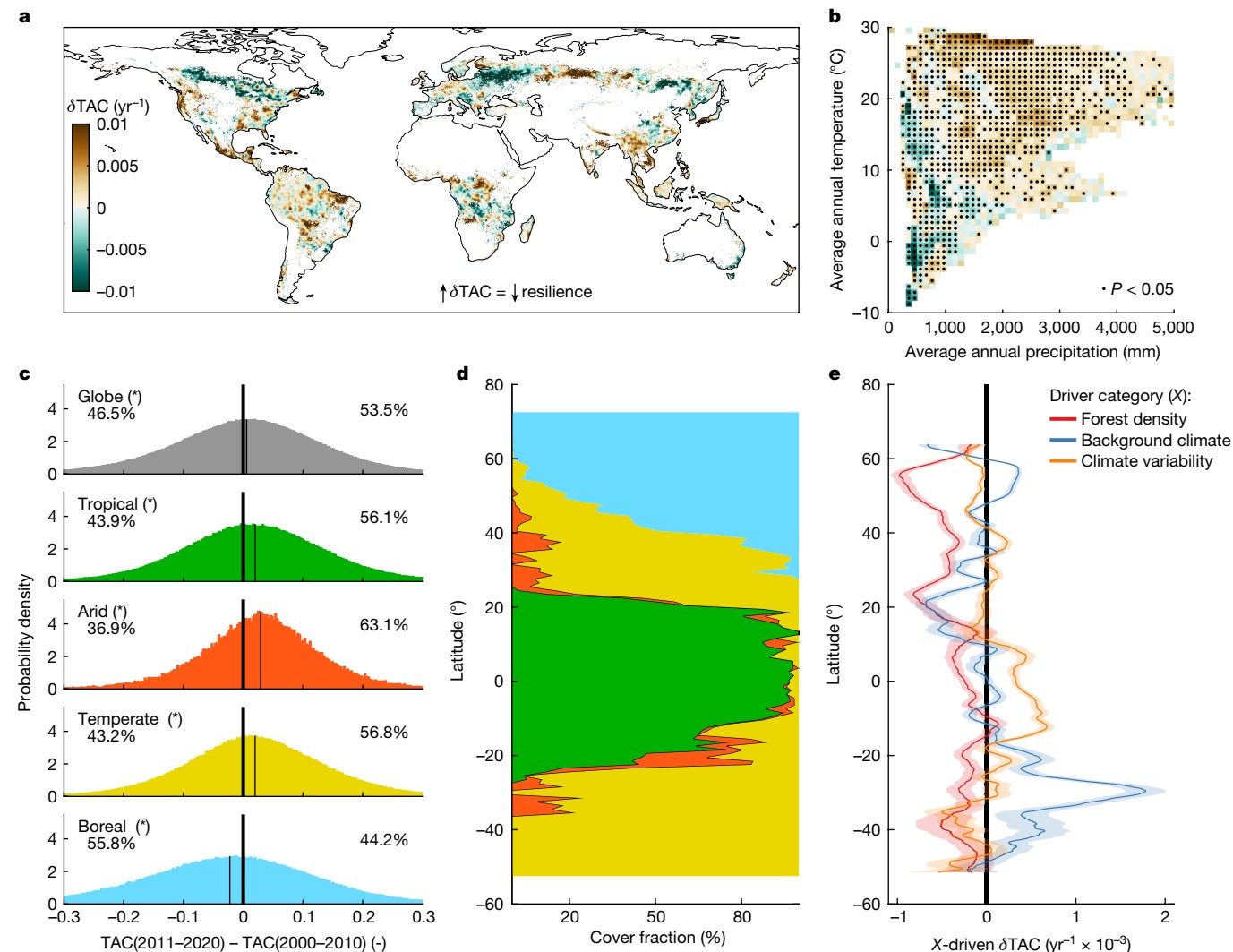

**Fig. 1 | Temporal variations of forest resilience and its key drivers. a**, Spatial map of the temporal trend of TAC ($\delta$TAC). Positive $\delta$TAC values (for example, tropical forests) suggest a reduction in recovery rates and thus a decline in resilience, and vice versa for negative $\delta$TAC values (for example, boreal forests). The values are averaged over a 1° × 1° moving window for visual purposes. **b**, $\delta$TAC as in **a** binned as a function of climatological temperature and precipitation. The black dots indicate bins with average values that are statistically different from zero (two-sided Student's $t$-test; $P$ value ≤ 0.05). **c**, Frequency distribution of the differences in TAC computed for two independent temporal windows (2011–2020 minus 2000–2010) and shown separately for different climate regions. The numbers refer to the percentage of the observations lower and greater than zero; the asterisks indicate distributions with averages that are statistically different from zero (two-sided Student's $t$-test; $P$ value ≤ 0.05). The thin vertical line in each plot shows the distribution average. **d**, The cover fraction corresponding to each climate region and colour code reported in **c** and shown over the latitudinal gradient. **e**, The zonal average of the trend in TAC ($\delta$TAC) as determined by the three drivers ($X$) at 5° latitudinal resolution and the corresponding 95% confidence interval shown as a coloured line and shaded band, respectively. The colours reflect the three different driver categories: forest density, background climate and climate variability.

## Resilience and primary productivity

Regardless of the forest type, changes in forest resilience may trigger variations in gross primary productivity (GPP) and vice versa, based on a mutual causal link. Understanding the interplay between these two variables is crucial given the role of GPP in the global carbon cycle[35]. We explored this by analysing the correlation of satellite-based GPP retrievals[36] and TAC at the annual scale (short-term interplay) and comparing the trends in GPP and TAC (long-term interplay; Methods). In the short term, intact forests show a lower correlation between GPP and TAC than managed forests (Fig. 2c), probably because resilience is on average higher in intact ecosystems (Fig. 2a) and therefore probably less critical for productivity. Such bi-directional interactions translate into a negative correlation between GPP and TAC, with a closer link in dry and cold climates, probably reflecting the potential amplification of the two-way

interplay in these regions (Fig. 2d,e). In the long term, about 70% of both managed and intact forests are experiencing a positive trend in GPP at present, but in 50% of these areas (about 36% in absolute terms), this occurs in combination with a positive trend in TAC (Fig. 2f, dark red patterns). This implies that a considerable fraction of forest area is increasing primary productivity while also experiencing a declining resilience, therefore leading to an expanding but more vulnerable forest sink. The widespread observations of rising tree mortality[2], as well as observations of the growing terrestrial carbon sink[37], confirm the co-occurrence of such antagonistic processes in response to global change[2].

## Early signals of abrupt forest decline

As a loss of forest resilience increases the sensitivity to external perturbations[14], we explored the potential of $\delta$TAC to work as an early-warning

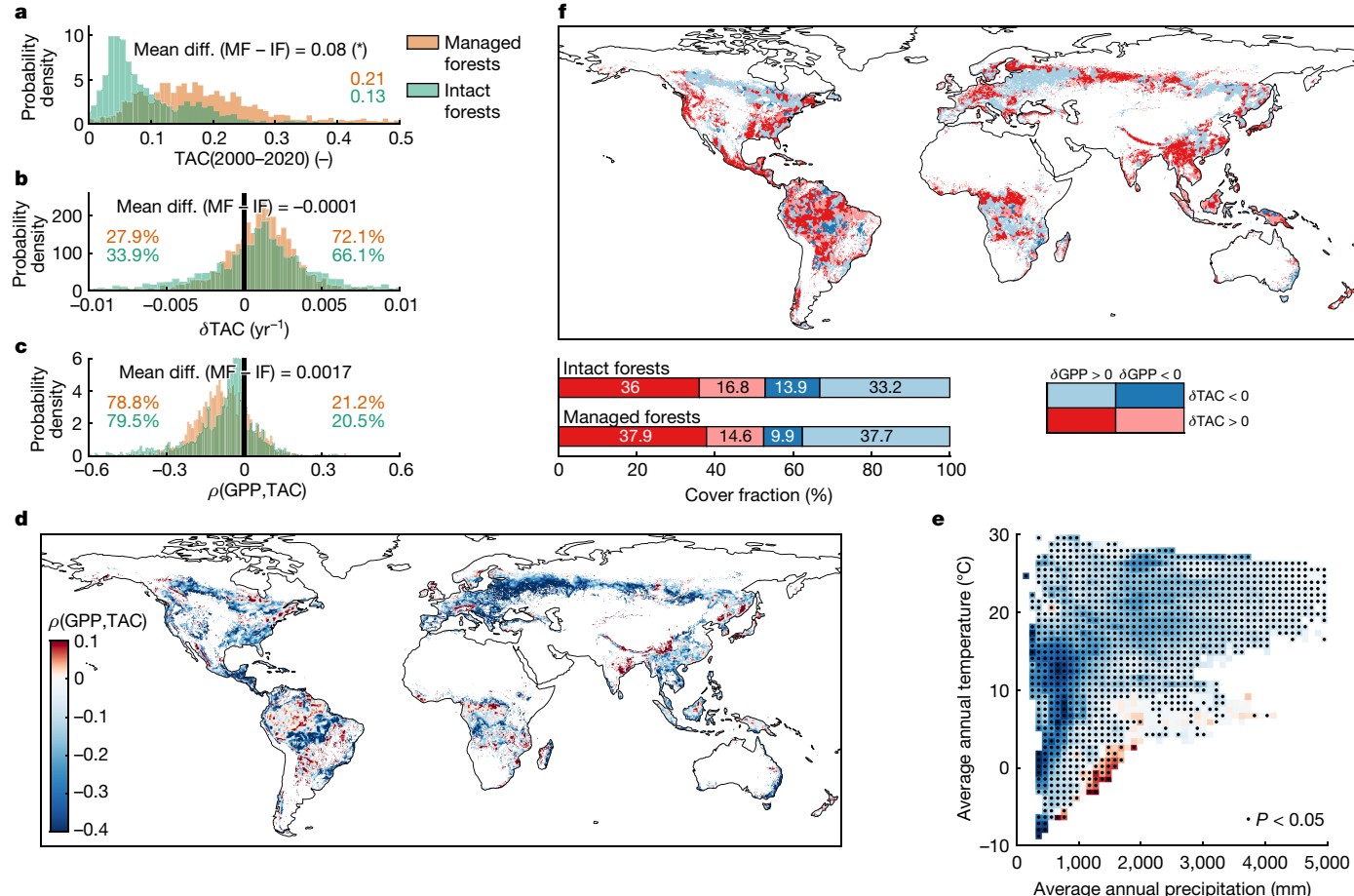

**Fig. 2 | Effect of forest management on forest resilience and interplay with GPP. a**, Frequency distributions of long-term TAC(2000–2020) for managed forests (MF) and intact forests (IF) located in a similar background climate. The coloured numbers report the respective averages, the top labels refer to the mean of the differences (diff.) in long-term TAC between managed and intact forests, and the asterisk indicates distributions that are statistically different (two-sided Student's *t*-test; *P* value ≤ 0.05). **b**, The same as for **a** but for δTAC; the coloured numbers refer to the percentage of the observations lower and greater than zero (on the left and right of 0 on the *x*-axis, respectively).

**c**, The same as for **b** but for the temporal correlation between annual GPP and TAC, denoted as ρ(GPP,TAC). **d**, A spatial map of ρ(GPP,TAC). **e**, ρ(GPP,TAC) binned as a function of climatological precipitation and temperature. The black dots indicate bins with average values that are statistically different from zero (two-sided Student's *t*-test; *P* value ≤ 0.05). **f**, A spatial map of the areas, with different colours for the four combinations of positive/negative δGPP and δTAC. The cover fractions of each of the four classes for managed and intact forests are reported in stacked bars.

signal of abrupt forest decline (theoretical framework described in Supplementary Methods 1 and 2). To exclude the effect of land management (for example, apparent abrupt declines (ADs) driven by forest harvest), we limited the analysis to global intact forests, with a focus on tropical and boreal regions that together cover about 97% of the investigated domain (Extended Data Fig. 8). ADs are defined here as sudden changes in the state of the forest ecosystem and detected, for a range of severities, as negative anomalies of 1 to 6 times the standard deviation (σ) of mean growing-season kNDVI with respect to the reference undisturbed mean in the time series. In this analysis, we quantify whether declining trends in resilience (that is, increases of δTAC) are associated with a consequent abrupt shift in the system, regardless of the disturbance type (details in Methods).

At the global level, intact forests have a probability of AD conditional on δTAC greater than 0.5 (Fig. 3a). This signal is statistically significant and increases with the severity of AD, suggesting that the progressive deterioration of ecosystem states, as tracked by the decline of resilience, has probably contributed to the upsurge of negative anomalies in forest dynamics. The emerging relation is mainly driven by boreal forests, particularly those in central Russia and western Canada, where there is an emergent, localized decline in forest resilience (Fig. 1a). Such patterns may indicate that in these zones the AD is following the drifting

towards a critical resilience threshold, which is probably triggered by the changes in environmental drivers occurring at the northernmost latitudes[38]. Insect outbreaks, which are typically favoured by water stress[39], may represent one of the main disturbances that have ultimately caused such ADs in the ecosystem state[40,41]. On the contrary, ADs in tropical forests are not statistically associated with high δTAC values (Fig. 3a). In these regions, fast and strong disturbance events, such as fires[42] or droughts[43], may induce an AD independently of long-term increasing trends in CSD (refs. [3,4]; here represented by δTAC). The above-mentioned hypotheses are also consistent with the dominant climate drivers of δTAC in boreal and tropical regions (background climate and climate variability, respectively, Fig. 1d,e) and further supported by several independent pieces of evidence (for example, refs. [26,30,44,45]).

## Critical threshold mechanisms

To further explore the threshold mechanisms and the causality associated with ADs, we retrieved TAC for the year preceding the occurrence of an AD (hereafter referred to as observed TAC_{AD})—and thus reflecting the threshold value of resilience before the AD of the ecosystem. For each AD event, we retrieved the corresponding ecosystem tolerance expressed as the difference between TAC_{AD} and its average TAC

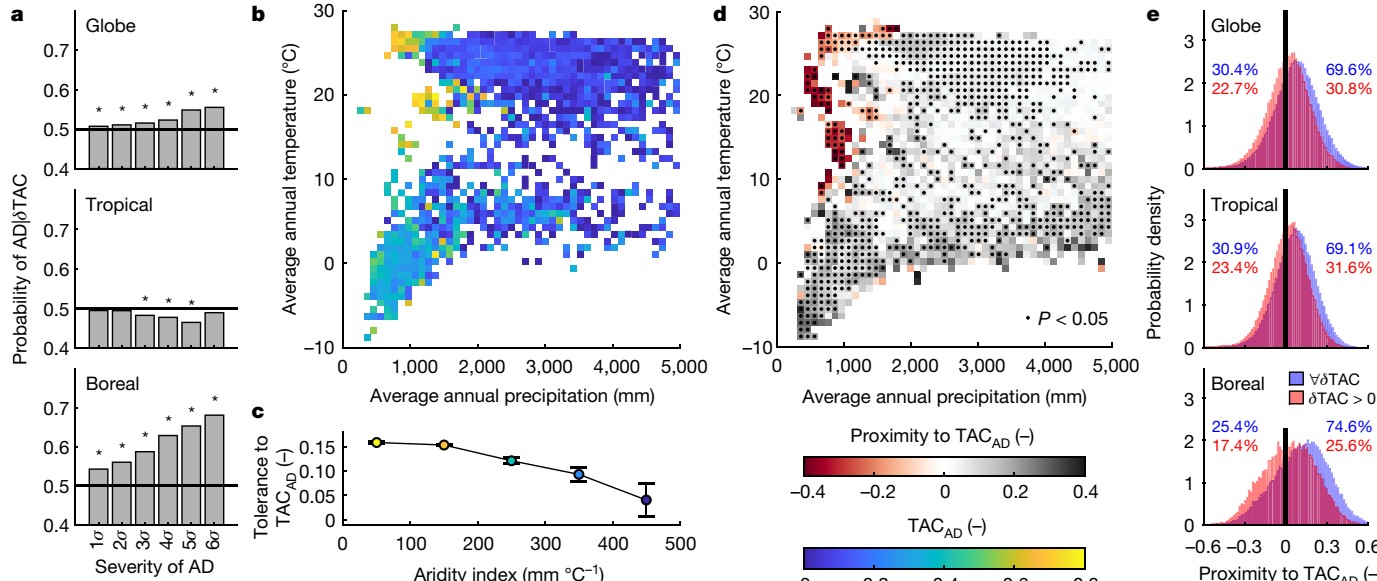

**Fig. 3 | Early-warning signals of ADs in intact forests. a**, Probability of occurrence of AD conditional on the values of $\delta$TAC for different severities of AD (expressed as anomaly $n$-times local standard deviation below the local mean, $\sigma$) shown separately for three different climate regions. The asterisks indicate probabilities statistically different from 0.5 (two-sided Student's $t$-test; $P$ value ≤ 0.05). **b**, TAC retrieved in the year preceding the occurrence of an AD (TAC_AD) binned as a function of climatological precipitation and temperature. **c**, Tolerance to TAC_AD (the absolute increase in TAC that an ecosystem in equilibrium can tolerate before reaching critical conditions) across a gradient of aridity index. The circle and whiskers refer to the average value and its 95% confidence interval; colours refer to the corresponding TAC_AD. Each binned aridity index ranging from 0 to 500 mm °C$^{-1}$ counts 10,868,

16,799, 728, 59 and 13 sampled pixels. **d**, Proximity to TAC_AD (proximity of present intact forests to their critical condition threshold) binned as a function of climatological precipitation and temperature. The black dots indicate bins with average values that are statistically different from zero (two-sided Student's $t$-test; $P$ value ≤ 0.05). Negative values of proximity to TAC_AD represent areas where the threshold resilience for AD (TAC_AD) has been already overpassed, and vice versa for positive values. **e**, Frequency distributions of proximity to TAC_AD shown separately for different climate regions and computed over the whole domain (blue) and over those areas experiencing a concomitant positive $\delta$TAC (red). The coloured numbers refer to the percentage of the frequency distribution lower and greater than zero (on the left and right of 0 on the $x$-axis, respectively) with respect to the whole domain.

computed in pre-disturbance conditions (details in Methods). This metric reflects the absolute increase in TAC that an ecosystem in equilibrium can tolerate before reaching critical conditions of AD. We found that, despite the average slow recovery rates (Extended Data Fig. 1 and Supplementary Discussion 1), ecosystems frequently exposed to water limitations experience ADs at higher levels of TAC_AD (Fig. 3b), thanks to their higher tolerance compared to tropical–humid and cold–dry forests (Fig. 3c). These patterns are probably due to the long-term adaptation of tree species in arid regions that leads to structural and physiological adaptation to water limitations (for example, deeper rooting systems, resistance to cavitation and higher root/shoot ratio), whereas humid and cold biomes have a higher vulnerability to water shortage[46,47].

To evaluate the proximity of present intact forests to their critical resilience threshold, we extrapolated in space the value of TAC_AD by the use of the RF regression algorithm and compared it with TAC retrieved for the year 2020. Proximity takes negative or zero values when TAC_AD has already been reached in 2020 and positive values when there are still margins before reaching the critical threshold (Methods). Results show that, at the end of our observational period, about 30% of global intact forests have already reached or overpassed their TAC_AD (Fig. 3d,e). More critically, about 23% experienced a concomitant increase in $\delta$TAC (Fig. 1a), therefore implying an ongoing reduction in ecosystem resilience to levels that are already close to an AD and, potentially, to a tipping point. We estimated that 3.32 Pg C of GPP is exposed to such critical conditions, prominently in tropical forests (93%), an amount about three times larger than the carbon losses due to deforestation in the Brazilian Amazon during the past ten years[26]. We point out that these critical conditions are not sufficient to determine a regime shift (Supplementary Methods 3). However, they represent a strong indication of

the rising risks of an increased instability and vulnerability to hazards of forest biomes. This is particularly critical for tropical forests, where the observed recent decline of the carbon sink[48,49] could by further exacerbated by the continuous and progressive deterioration of forest resilience and the parallel increase in tree mortality and turnover rate.

## Conclusions

Our analysis reveals that in recent decades both intact and managed forests have experienced substantial changes in resilience controlled by large-scale climate signals. We found that tropical, temperate and arid forests underwent a decline in resilience probably related to the concomitant increase in water limitations and climate variability. On the contrary, benefits induced by climate warming and $CO_2$ fertilization have outweighed such negative effects in much of the boreal biome, ultimately leading to an increase in forest resilience. The increasing fragility to external perturbations in combination with an enhancement in productivity for a considerable fraction of global forests (about 36%) confirms the co-occurrence of antagonistic processes driving photosynthesis and tree mortality in response to global change[2]. We estimate that about 23% of intact undisturbed forests have already reached their critical threshold for an AD and are experiencing a concomitant further degradation of resilience. Considering the expected transition from a $CO_2$-fertilization-dominated period to a warming/drying-dominated period[27–29], the observed negative trajectories of forest resilience suggest potential critical consequences for key ecosystem services, such as carbon sequestration. Therefore, it is becoming urgent to account for these trends in the design of effective forest-based mitigation strategies to avoid future unexpected negative events triggered by the increasing vulnerability of carbon stocks. In this regard,

our global data-driven assessment shows that resilience thinking[50] can be developed effectively in a science-based and solution-oriented framework to support the many challenges of forest management in times of rapid climatic changes.

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

## Methods

### Climate drivers

To explore the impact of climate on forest resilience (see the following sections), we used monthly averaged total precipitation, 2-m air temperature, evapotranspiration deficit and surface solar radiation downwards acquired from the ERA5-Land reanalysis product at 0.1° spatial resolution for the 2000–2020 period (https://cds.climate.copernicus.eu/cdsapp#!/home). Evapotranspiration deficit was quantified as the total precipitation minus evapotranspiration. In this study, we referred to climate regions as defined by the Köppen–Geiger world map of climate classification[51] (http://koeppen-geiger.vu-wien.ac.at/present.htm). The original 31 climatic zones were merged into major zones and only those characterized by vegetation cover were included in our study (tropical, arid, temperate and boreal; Extended Data Fig. 8).

### Vegetation dynamics

NDVI data acquired from the Moderate Resolution Imaging Spectroradiometer (MODIS) instrument aboard the Terra satellite was used to derive changes in global vegetation for the period 2000–2020. We used cloud-free spatial composites provided at 16-day temporal resolution and 0.05° spatial resolution (MOD13C1 Version 6; https://lpdaac.usgs.gov/products/mod13c1v006/) and retained only pixels with good and marginal overall quality. The MODIS-derived NDVI dataset represents a state-of-the-art product of vegetation state whose retrieval algorithm is constantly improved[52], and being derived from a unique platform and sensor, it is temporally and spatially consistent. Vegetation dynamics were analysed in terms of kNDVI, a nonlinear generalization of the NDVI based on ref. [22] and derived as follows:

$$kNDVI = \tanh(NDVI^2) \tag{1}$$

kNDVI has recently been proposed as a strong proxy for ecosystem productivity that shows high correlations with both plot level measurements of primary productivity and satellite retrievals of sun-induced fluorescence[22]. In addition, kNDVI has been documented to be more closely related to primary productivity, to be resistant to saturation, bias and complex phenological cycles, and to show enhanced robustness to noise and stability across spatial and temporal scales compared to alternative products (for example, NDVI and near-infrared reflectance of vegetation). For these reasons, it has been retained in this study as the preferred metric to describe the state of the forest ecosystem.

To obtain an accurate estimate of resilience indicators, vegetation time series need to be stationary without seasonal periodic patterns or long-term trends[53]. To this aim, vegetation anomalies were obtained from kNDVI data by first subtracting the multi-year 16-day sample mean and then removing linear trends from the resulting time series. Missing data, due for instance to snow cover affecting the retrieval of reflectance properties, have been gap-filled by climatological kNDVI values. The time series of kNDVI-based vegetation anomalies was used to derive resilience indicators and assess their spatial and temporal variations (see next sections).

Interannual changes in vegetation were assessed in terms of growing-season-averaged kNDVI. To this end, a climatological growing season that spanned months with at least 75% of days in the greenness phase was derived from the Vegetation Index and Phenology satellite-based product[54] (https://vip.arizona.edu/) and acquired for the 2000–2016 period at 0.05° spatial resolution. In addition, forest cover (FC) fraction was derived from the annual land-cover maps of the European Space Agency's Climate Change Initiative (https://www.esa-landcover-cci.org/)[55] over the 2000–2018 period at 300-m spatial resolution. FC was retrieved by summing the fraction of broadleaved deciduous, broadleaved evergreen, needle leaf deciduous and needle leaf evergreen forest. FC was resampled to 0.05° to match the kNDVI spatial resolution.

### Spatial patterns of slowness and its dependence on environmental factors

In this study, we quantified the resilience of forest ecosystems—their ability to recover from external perturbations—by the use of the 1-lag TAC (refs. [3–5]). Such an indicator was initially computed on the whole time series of vegetation anomalies (2000–2020) for forest pixels with less than 50% missing data in the original NDVI and FC greater than 0.05 and referred to in the text as long-term TAC. This analysis was used to assess the spatial patterns of the forest slowness mediated by environmental factors that affect plant growth rates and capacity to recover from perturbations. The long-term TAC was explored both in the geographic and climate space (Extended Data Fig. 1). In the climate space, long-term TAC was binned in a 50 × 50 grid as a function of average annual precipitation and temperature, both computed over the 2000–2020 period, using the average as an aggregation metric weighted by the areal extents of each record. We retained only bins with at least 50 records.

To explore the potential drivers of long-term TAC, we developed an RF regression model[23] and predicted the observed long-term TAC (response variable) based on a set of environmental features (predictors). The use of machine learning in general and of RF in particular, being nonparametric and nonlinear data-driven methods, does not require a priori assumptions about the functional form relating the key drivers and the response functions. The environmental variables include vegetation properties (FC and growing-season-averaged kNDVI) and climate variables (total precipitation, 2-m air temperature, evapotranspiration deficit and surface solar radiation downwards). Each of the climate variables was expressed in terms of average, coefficient of variation and 1-lag autocorrelation and resampled to 0.05° spatial resolution to match the spatial resolution of kNDVI. All environmental variables were computed annually and then averaged over time, except the autocorrelation that was computed directly for the whole period, analogously to the long-term TAC. This resulted in a set of 14 predictors representing the forest density, the background climate, the climate variability and its TAC in the observational period (Extended Data Table 1). The RF model was developed by splitting the observed long-term TAC into two separate samples: 60% of records were used for model calibration, and the remaining 40% were used to validate model performances in terms of coefficient of determination ($R^2$), mean squared error and percentage bias (PBIAS). Each record refers to a 0.05° pixel. The RF implemented here uses 100 regression trees, whose depth and number of predictors to sample at each node were identified using Bayesian optimization. The general model formulation is as follows:

$$TAC = f(X) + \varepsilon_f \tag{2}$$

in which $f$ is the RF regression model, $X$ are the environmental predictors and $\varepsilon_f$ are the residuals. We found that the model explains 87% of the spatial variance ($R^2$) of the observed long-term TAC with a mean squared error of 0.007 and an average overestimation of 0.058 (PBIAS; Extended Data Fig. 2a). By definition, machine learning methods are not based on the mechanistic representation of the phenomena and therefore cannot provide direct information on the underlying processes influencing the system response to drivers. However, some model-agnostic methods can be applied to gain insights into the outputs of RF models. Here we used variable importance metrics to quantify and rank how individual environmental factors influence TAC (Extended Data Fig. 2b). Furthermore, using partial dependence plots derived from the machine learning algorithm RF, we explored the ecosystem response function (TAC) across gradients of vegetation and climate features (Supplementary Discussion 1 and Extended Data Fig. 2c–f).

## CSD indicators

To explore the temporal variation in forest resilience, we used CSD indicators, here quantified in terms of temporal changes in TAC retrieved for two consecutive and independent periods ranging from 2000 to 2010 and from 2011 to 2020, and assessed the significance of the change in the sampled mean aggregated for different climate regions through a two-sided $t$-test (Fig. 1c). This analysis was complemented by the computation of TAC on the annual scale over a 2-year lagged temporal window (3-year window size) to track the temporal changes in CSD. This resulted in a time series of TAC with an annual time step.

We point out that temporal dynamics of annual TAC are driven by two processes: the changes in the resilience of the system that affect the velocity of the recovery from external perturbations and the confounding effects of the changes in autocorrelation of the climate drivers ($X_{ac}$) that directly affect the autocorrelation of NDVI. Given the specific goals of this study, we factored out the second process from the total TAC signal to avoid that an increasing autocorrelation in the drivers would affect our analysis and conclusions about the resilience and the potential increase in instability[56]. For this purpose, we disentangled the temporal changes in TAC due to variations in autocorrelation in the climate drivers ($TAC|X_{ac}$) by adopting the space-for-time analogy and applied the RF model ($f$) at an annual time step ($t$) in a set of factorial simulations as follows:

$$TAC^t \,|\, X_{ac} = f(X^t) - f(X^t_{-ac}, X^{2000}_{ac}) \tag{3}$$

The first term on the right side of equation (3) is the RF model simulation obtained by accounting for the dynamics of all predictors, and the second term is the RF model simulation generated by considering all predictors dynamic except the factors of autocorrelation in climate that are kept constant to their first-year value (year 2000). For such runs, we used predictors computed on an annual scale over a 2-year lagged temporal window, consistently to the TAC time series. We found that the direct effects of autocorrelation in climate have led to a positive trend of TAC in dry zones (due to the increasing autocorrelation of the drivers in these regions) and to an opposite effect in temperate humid forests (Supplementary Fig. 3). To remove these confounding effects, the estimated term $TAC^t|X_{ac}$ is factored out from the $TAC^t$ by subtraction to derive an enhanced estimate of annual resilience that is independent of autocorrelation in climate (Extended Data Fig. 3).

Long-term linear trends computed on the resulting enhanced TAC time series ($\delta TAC$) represent our reference CSD indicator used in this study to explore the changes in forest resilience. $\delta TAC$ was quantified for each grid cell (Fig. 1a) and represented in the climate space following the methodology previously described (Fig. 1b). We then assessed the significance of the trends at bin level by applying a two-sided $t$-test for the sampled trend distributions within each bin. This significance test is independent from the structural temporal dependencies originating from the use of a 2-year lagged temporal window to compute the TAC time series.

Following an analogous approach described in equation (3), we disentangled the effect of the variation in forest density, background climate and climate variability on temporal changes in TAC (Fig. 1d,e). We recognize that other environmental factors not explicitly accounted for in our RF model could play a role in modulating the temporal variations in TAC. However, given the comprehensiveness of the suite of predictors used in equation (2) (Extended Data Table 1), it seems plausible that residuals mostly reflect the intrinsic forest resilience, the component intimately connected to the short-term responses of forests to perturbations, which is not directly related to climate variability. Forest ecosystem evolutionary processes could also play a role, but longer time series would be required to reliably capture these dynamics. Furthermore, abrupt declines (ADs) in the vegetation state and following recoveries, similarly to those potentially originating from forest disturbances (for example, wildfires and insect outbreaks), could influence the TAC changes. However, such occurrences, being distributed across the globe throughout the whole period, are expected to only marginally affect the resulting trend in TAC time series.

## Sensitivity analysis

To assess the robustness of our results with respect to the modelling choices described above, we performed a series of sensitivity analyses for the difference in TAC retrieved for the two independent periods (2000–2010 and 2011–2020). To this aim, we tested their dependence on: the quality flag of the NDVI data used for the analyses (good, good and marginal); the gap-filling procedure tested on different periods (year and growing season); the inclusion or exclusion of forest areas affected by ADs; the threshold on the maximum percentage of missing NDVI data allowed at the pixel level (20%, 50% and 80%); the threshold on the minimum percentage of FC allowed at the pixel level (5%, 50% and 90%); and the pixel spatial resolution used for the analyses (0.05°, 0.25° and 1°). In addition, we tested the sensitivity of the trend in total TAC signal on the moving temporal window length used to calculate autocorrelation at lag 1. Results obtained for the different configurations were compared in terms of frequency distributions, separately for climate regions (Extended Data Fig. 4), and further explored in the climate space (Extended Data Figs. 5 and 6). Outcomes of the sensitivity analysis are discussed in Supplementary Discussion 2.

## Interplay between GPP and CSD

Resilience and GPP interact with each other through mutual causal links. On one hand, a reduction in forest resilience makes the system more sensitive to perturbations with potential consequent losses in GPP (ref. [26]). On the other hand, a reduction in GPP may lead to a decline in resilience according to the carbon starvation hypothesis, and may be associated with increasing hydraulic failure[46]. To explore the link between forest resilience and primary productivity, we quantified the correlation between TAC and GPP. Estimates of GPP were derived from the FluxCom Model Tree Ensemble for the 2001–2019 period at 8-daily temporal resolution and 0.0833° spatial resolution and generated using ecosystem GPP fluxes from the FLUXNET network and MODIS remote sensing data as predictor variables[36] (http://www.fluxcom.org/). Annual maps of GPP were quantified and resampled to 0.05° to match the temporal and spatial resolution of TAC time series. The Spearman rank correlation ($\rho$) was then computed between annual GPP and TAC over a 1° spatial moving window to better sample the empirical distribution of the two variables (Fig. 2d). The significance of $\rho$(GPP,TAC) was assessed over the climate space separately for each bin (Fig. 2e), similarly to the approach used to test the significance of $\delta TAC$. Furthermore, we explored the relationships between the trend in GPP ($\delta GPP$) and the trend in TAC ($\delta TAC$) by clustering the globe according to the directions of the long-term trajectories of the above-mentioned variables (Fig. 2f).

## Disentangling the impact of forest management

To characterize TAC on different forest types and disentangle the potential effects originating from forest management, results were separately analysed for intact forests and managed forests. Intact forests were considered those forest pixels constituting the Intact Forest Landscapes[57] dataset (https://intactforests.org/). Intact Forest Landscapes identifies the forest extents with no sign of significant human activity over the period 2000–2016 based on Landsat time series. The remaining forests pixels—not labelled as intact—were considered as managed forests (Extended Data Fig. 8). The resulting forest type map is consistent with those used for United Nations Framework Convention on Climate Change reporting[58], although with more conservative estimates of intact forests in the boreal zone due to the masking based on FC and percentage of missing data applied in this study.

We analysed the differences in long-term TAC (computed for the whole 2000–2020 period) between managed and intact forests by masking out the potential effect of climate background. To this aim, we compared the climate spaces generated separately for managed and intact forests by extracting only those bins that are covered by both forest classes. The resulting distributions—one for each forest class—have the same sample size, and each pair of elements shares the same climate background. Potential confounding environmental effects on average recovery rates are, therefore, minimized. We then applied a two-sided $t$-test for analysing the significance of the difference in the sampled means (Fig. 2a). An analogous approach was used to test the differences in $\delta$TAC and $\rho$(GPP,TAC) between managed and intact forests (Fig. 2b,c).

### Early-warning signals of abrupt forest declines

When forest ecosystems are subject to an extended and progressive degradation, the loss of resilience can lead to an AD (refs. [3–5]). Such abrupt changes can trigger a regime shift (tipping point) depending on the capacity of the system to recover from the perturbations (Supplementary Methods 1 and 2). We investigated the potential of changes in TAC as early-warning signals of ADs in intact forests over the 2010–2020 period. To this aim, we quantified at the pixel level ADs as the events occurring on a certain year when the corresponding growing-season average kNDVI was more than $n$-times local standard deviation below the local mean. Local mean and standard deviation ($\sigma$) were computed over the 10-year antecedent temporal window (undisturbed) period and $n$ varies between 1 and 6 with higher values reflecting more severe changes in the state of the system. For each pixel and for each fixed $n$ value, we recorded only the first AD occurrence, thus imposing a univocal record for each abrupt change in the state of the system.

We then explored whether the retrieved ADs were statistically associated with antecedent high values of $\delta$TAC. To avoid confusion with the attribution of causality, for each AD that occurred at time $t$ (over the 2010–2020 period), we derived the $\delta$TAC over the temporal window $2000 - (t - 1)$. The resulting trend in $\delta$TAC is therefore antecedent and independent of the changes in vegetation associated with the AD. Then, for each pixel with an AD at time $t$, we also extracted randomly one of the undisturbed (with no AD) adjacent pixels and retrieved $\delta$TAC over the same temporal window. This analysis produced two distributions of $\delta$TAC associated with pixels with and without ADs (AD and no AD, respectively). The two distributions have the same size and each pair of elements shares similar background climate. We calculated the probability of occurrence of AD conditional on the trend in $\delta$TAC (AD|$\delta$TAC) as the frequency of ADs for which $\delta$TAC (AD)| > $\delta$TAC (no AD), and the significance of the difference in the two sampled means (AD and no AD) was evaluated through a two-sided $t$-test. Probability and significance were assessed for different climate regions and severity of ADs (Fig. 3a). High statistically significant probabilities suggest that the AD is following the drifting towards a critical resilience threshold plausibly associated with changes in environmental drivers.

We complemented the aforementioned analyses by retrieving the tolerance and proximity to AD, hereafter determined for a 3$\sigma$ severity. We first quantified the TAC that proceeded the occurrence of an AD and followed a progressive loss of resilience as captured by positive $\delta$TAC. This value, hereafter referred to as abrupt decline temporal autocorrelation ($TAC_{AD}$), reflects the TAC threshold over which we observed an abrupt change in the forest state (Fig. 3b). The tolerance to AD was quantified as the difference between the local $TAC_{AD}$ and the TAC value averaged over the 2000–2009 period to characterize the pre-disturbance conditions. The tolerance metric was explored across a gradient of aridity index[59] (Fig. 3c).

$TAC_{AD}$ can be directly retrieved only on those forest pixels that have already experienced an AD. As a considerable fraction of undisturbed forests could potentially be close to their critical TAC threshold, or even have already passed it, it is important to determine their $TAC_{AD}$. To this

aim, we developed an RF regression model that expresses the $TAC_{AD}$ as a function of the set $X$ of environmental variables used in model $f$ (equation (2)) but excluding the autocorrelation in climate drivers ($X_{reduced}$) already disentangled in the TAC signal. The general formulation is as follows:

$$TAC_{AD} = g\,(X_{reduced}) + \varepsilon_g \qquad (4)$$

in which $g$ is the RF regression model, $X_{reduced}$ are the environmental predictors and $\varepsilon_g$ are the residuals. Implementation, calibration and validation of $g$ follow the same rationale described before for the $f$ model. We found that the RF model explains 50% of the variance ($R2$) of the observed $TAC_{AD}$, with a mean squared error of 0.019 and an average underestimation of 0.86 (PBIAS).

The RF model was then used to predict the $TAC_{AD}$ over the whole domain of intact forests and served as input to quantify the proximity to AD of undisturbed forest pixels at the end of the observational period (year 2020). Here we defined the proximity metric as the difference between the value of TAC in 2020 and $TAC_{AD}$. Proximity takes negative or zero values when $TAC_{AD}$ has already been reached ($TAC^{2020} \geq TAC_{AD}$) and positive values when there are still margins before reaching the critical threshold ($TAC^{2020} < TAC_{AD}$). Together $\delta$TAC > 0 and $TAC^{2020} \geq TAC_{AD}$ therefore represent the most critical conditions, as they indicate that the critical resilience threshold for AD has already been reached and the ecosystem is continuing to lose its capacity to respond to external perturbations. We finally quantified the amount of GPP potentially exposed to such critical conditions by linearly extrapolating the GPP for the year 2020 (available GPP data stop in 2019) and overlaying it on the map of critical conditions (proximity to AD < 0 and $\delta$TAC > 0).

### Reporting summary

Further information on research design is available in the Nature Research Reporting Summary linked to this paper.

### Data availability

The climate datasets used in this study are publicly available from the ERA5-Land reanalysis product (https://cds.climate.copernicus.eu/cdsapp#!/home) and from the Köppen–Geiger world map of climate classification (http://koeppen-geiger.vu-wien.ac.at/present.htm). NDVI data were acquired from MODIS (MOD13C1 Version 6, https://lpdaac.usgs.gov/products/mod13c1v006/), land surface phenology data were acquired from the Vegetation Index and Phenology satellite-based product (https://vip.arizona.edu/), and FC data were acquired from the European Space Agency's Climate Change Initiative (https://www.esa-landcover-cci.org/). GPP fluxes are available from the FluxCom product (http://www.fluxcom.org/) and the spatial delineation of intact forests is available from the Intact Forest Landscapes dataset (http://intactforests.org/).

### Code availability

The custom MATLAB (R2017b) code written to analyse the data, develop the RF model and generate figures is available at https://doi.org/10.6084/m9.figshare.19636059.v1.

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

**Acknowledgements** The study was funded by the Exploratory Project FOREST@RISK of the European Commission, Joint Research Centre. N.G.M. was supported by the Department of Energy's project Next Generation Ecosystem Experiment-Tropics.

**Author contributions** G.F. and A.C. designed the study; G.F. developed the analyses; G.F. and A.C. interpreted the results, G.F. wrote the manuscript with contributions from all coauthors.

**Competing interests** The authors declare no competing interests.

**Additional information**
**Correspondence and requests for materials** should be addressed to Giovanni Forzieri.

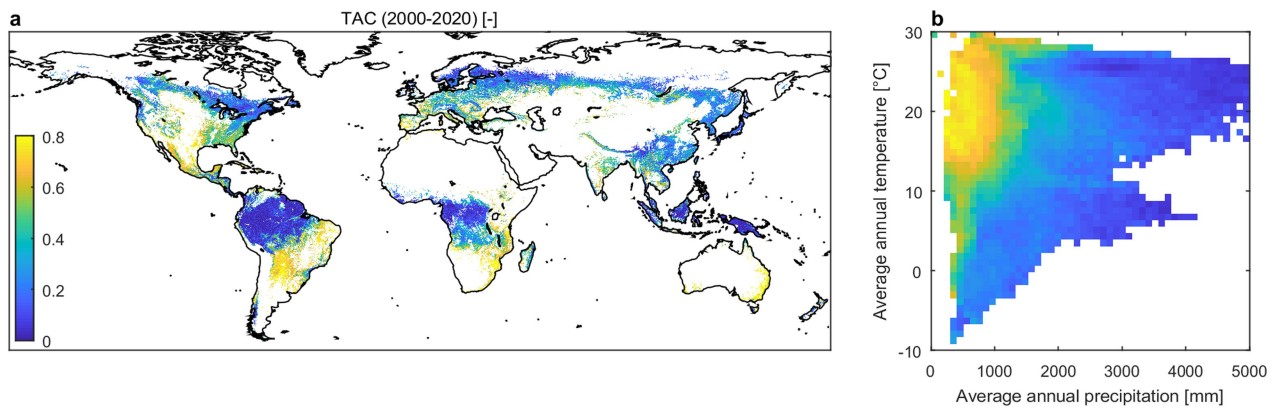

**Extended Data Fig. 1 | Spatial variation of forest slowness.** (**a**) Spatial map of long-term *TAC* computed for the whole 2000-2020 period. (**b**) Long-term *TAC* binned as a function of climatological temperature and precipitation.

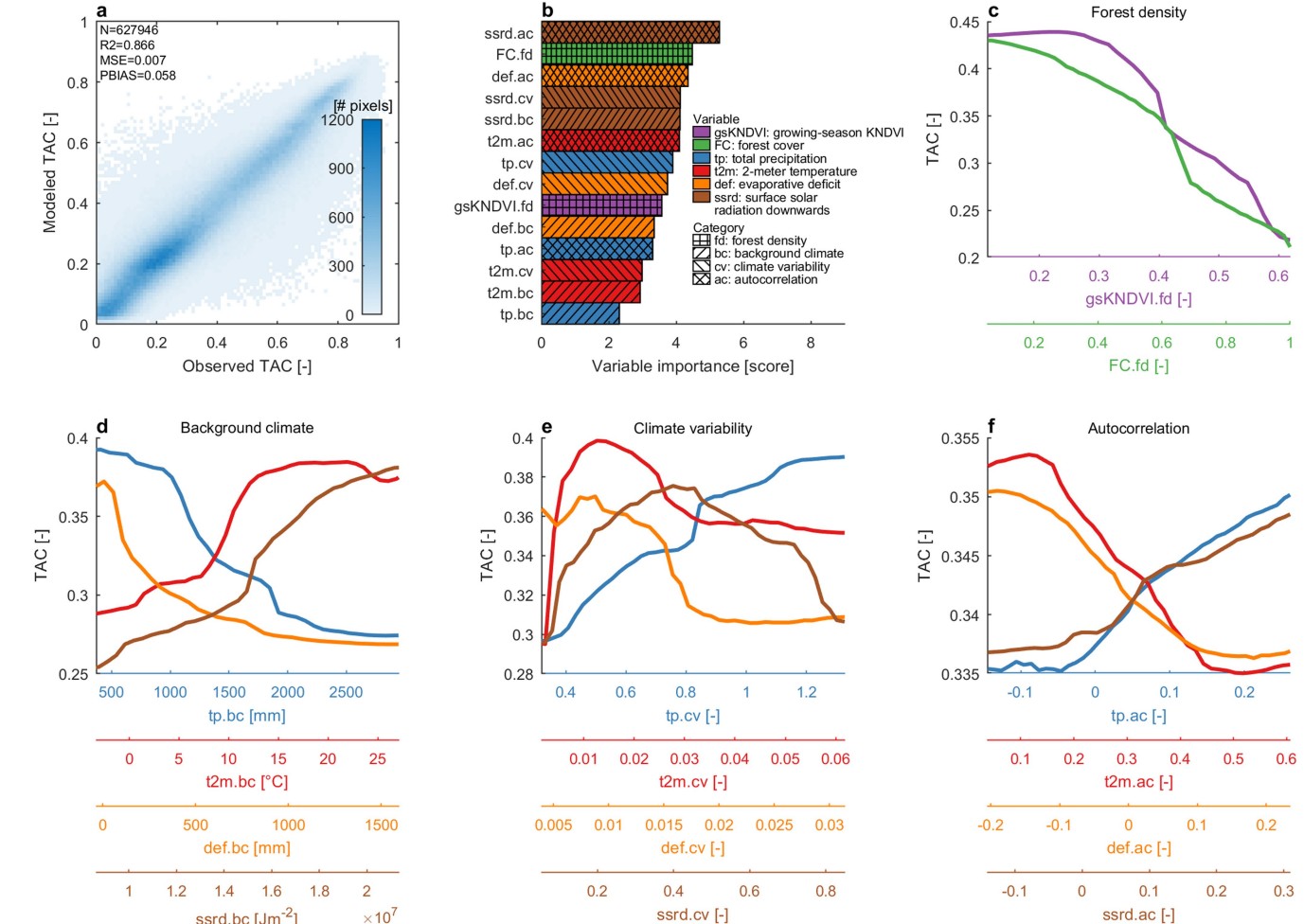

**Extended Data Fig. 2 | Performance and response functions of the resilience model.** (**a**) Observed versus modelled long-term *TAC*. Number of binned records (*N*), coefficient of determination (*R²*), mean squared error (MSE) and percent bias (PBIAS) are shown in labels, while the frequency distribution in color. (**b**) Predictors of *TAC* and corresponding variable importance based on the random forest regression model of forest resilience.

The four categories of environmental predictors are identified with hatched fill patterns; whereas the colors distinguish the different variables. (**c**) Dependence of *TAC* on predictors of forest density. (**d**), (**e**) and (**f**) as (**c**) but for predictors of background climate, climate variability and autocorrelation, respectively.

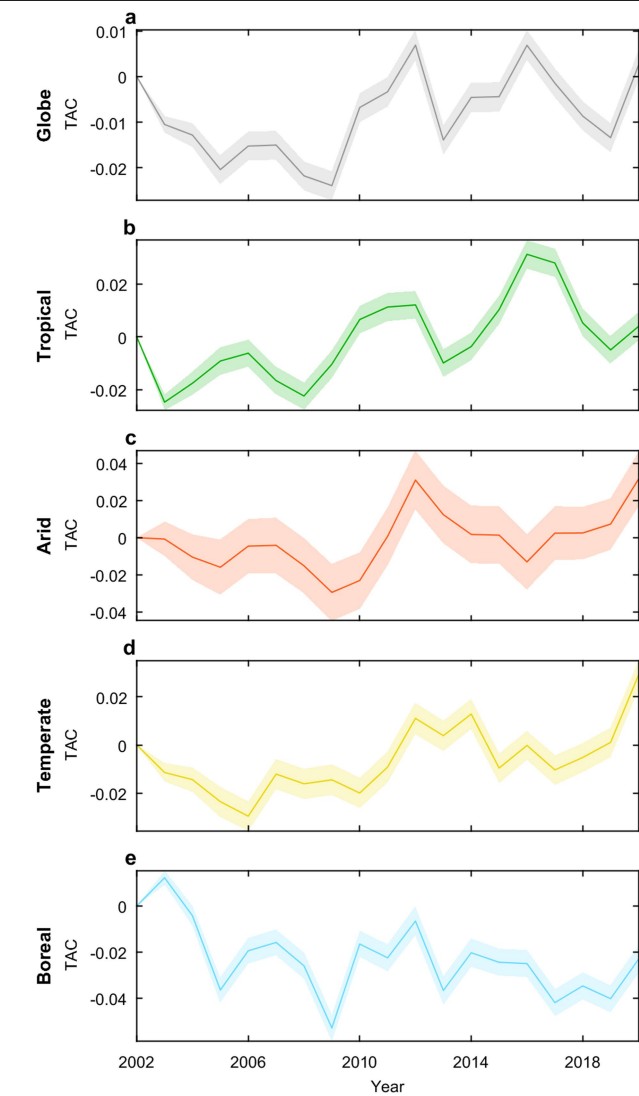

**Extended Data Fig. 3 | Temporal trajectories of forest resilience.** Temporal changes in *TAC* computed over a 3-year moving window and displayed with respect to the reference year 2002 separately for the global (**a**), tropical (**b**), arid (**c**), temperate (**d**) and boreal (**e**) regions. Continuous lines refer to the regional averages, whereas shaded areas show their 95% confidence interval magnified by a factor of 10 for visual purposes.

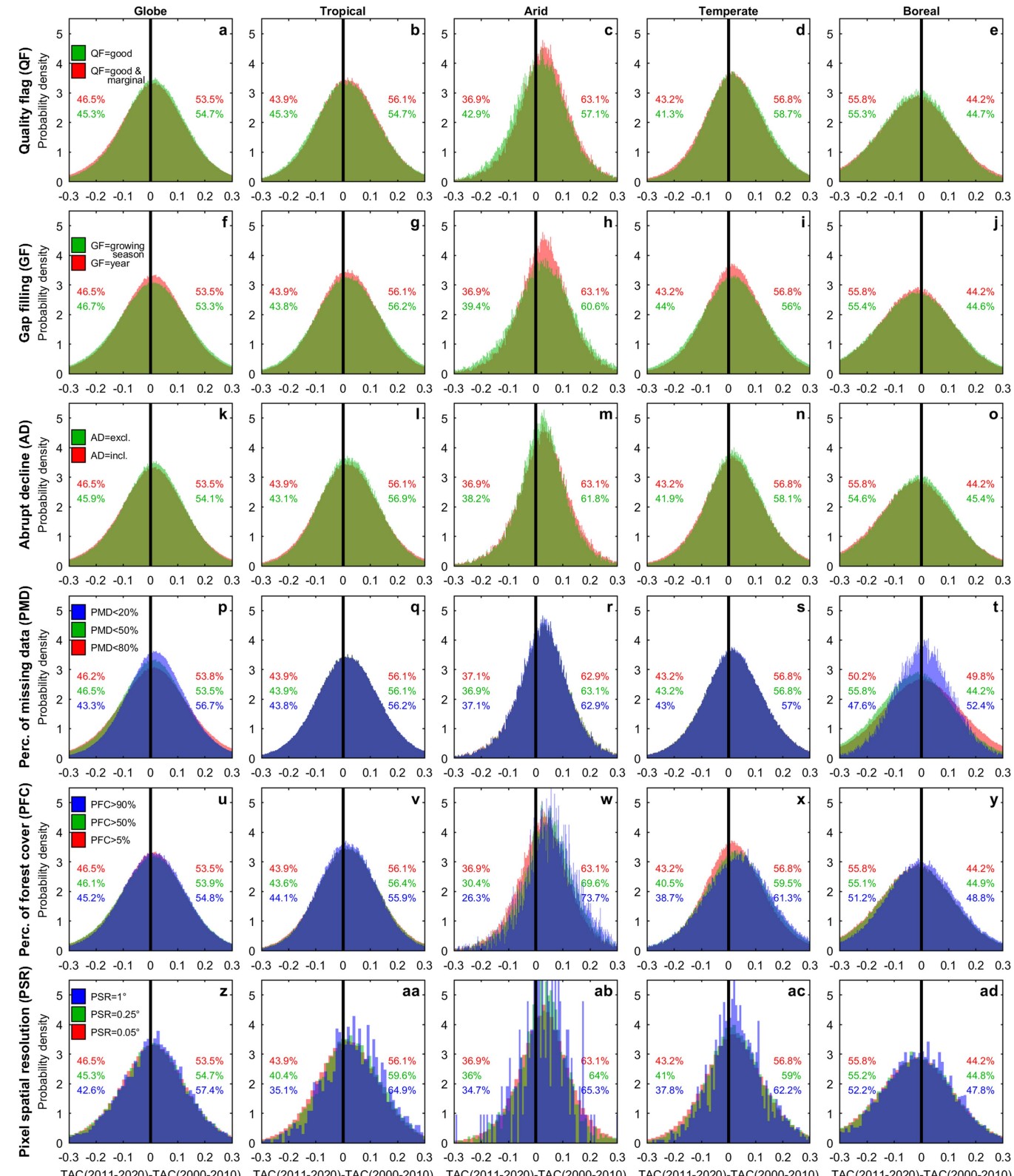

**Extended Data Fig. 4 | Sensitivity analysis of temporal changes in forest resilience (frequency distributions).** (**a**–**e**) Frequency distribution of the differences in *TAC* computed for two independent temporal windows (2011-2020 and 2000-2010) shown separately for different climate regions and for the use of different quality flags of NDVI data (QF). Numbers refer to the percentage of the frequency distribution lower and greater than zero (on the left and right y-axis, respectively). (**f**–**j**), (**k**–**o**), (**p**–**t**), (**u**–**y**) and (**z**–a**d**) as (**a**–**e**) but computed for different gap filling analyses (GF), inclusion/exclusion of areas affected by abrupt declines (AD), percentages of missing data (PMD), percentages of forest cover (PFC) and spatial resolution (PSR), respectively.

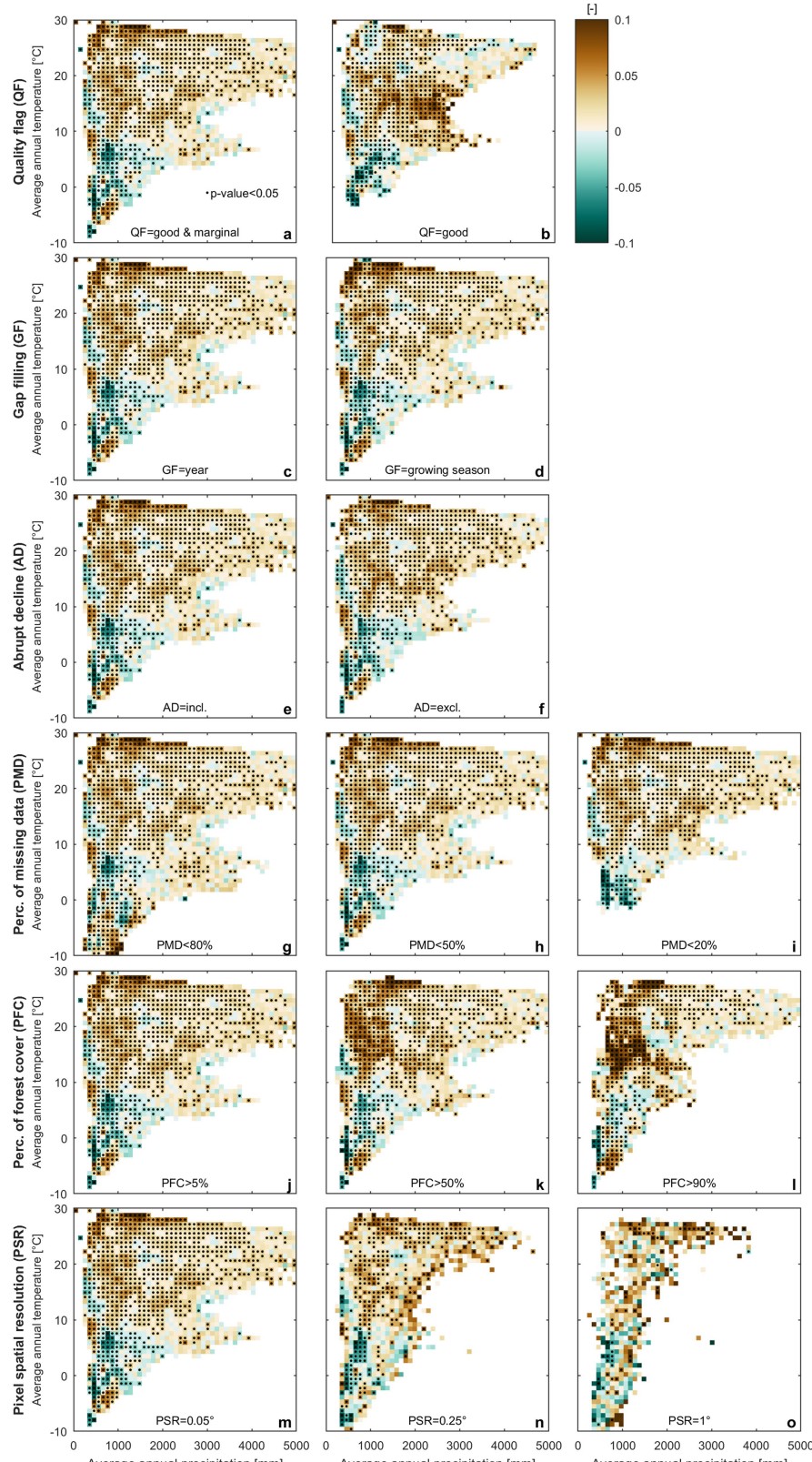

**Extended Data Fig. 5 | Sensitivity analysis of temporal changes in forest resilience (climate spaces). (a–b)** Differences in *TAC* computed for two independent temporal windows (2011-2020 and 2000-2010), separately shown for different quality flags (QF), binned as a function of climatological temperature and precipitation. Black dots indicate bins with average values that are statistically different from zero (two-sided Student's *t*-test; *P*-value ≤ 0.05). **(c–d)**, **(e–f)**, **(g–i)**, **(j–l)** and **(m–o)** as **(a–b)** but computed for different gap filling analyses (GF), inclusion/exclusion of areas affected by abrupt declines (AD), percentages of missing data (PMD), percentages of forest cover (PFC) and spatial resolution (PSR), respectively.

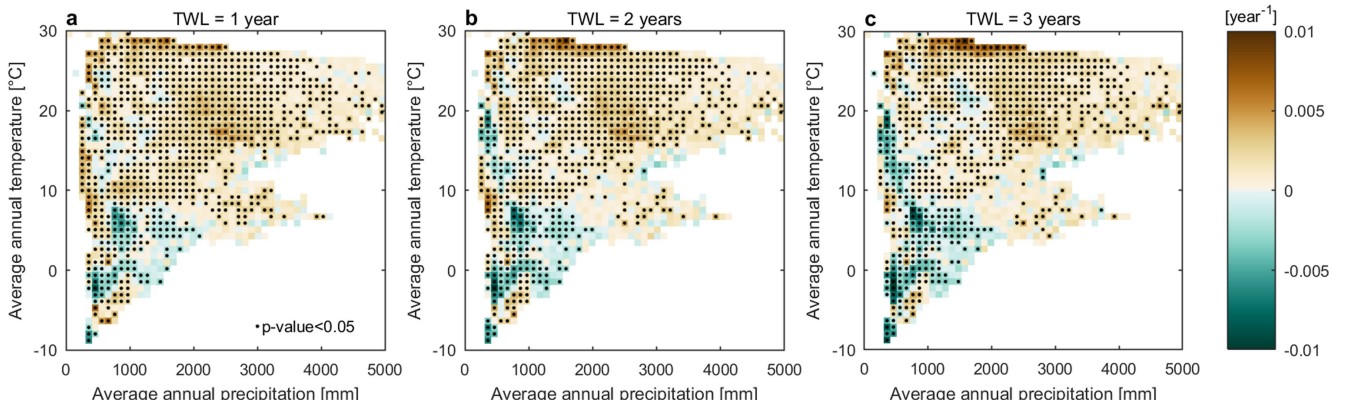

**Extended Data Fig. 6 | Effects of varying lagged temporal windows lengths.** (**a**–**c**) Trend in total *TAC* binned as a function of climatological temperature and precipitation, separately shown for different temporal window lengths (TWL). Black dots indicate bins with average values that are statistically different from zero (two-sided Student's *t*-test; *P*-value ≤ 0.05).

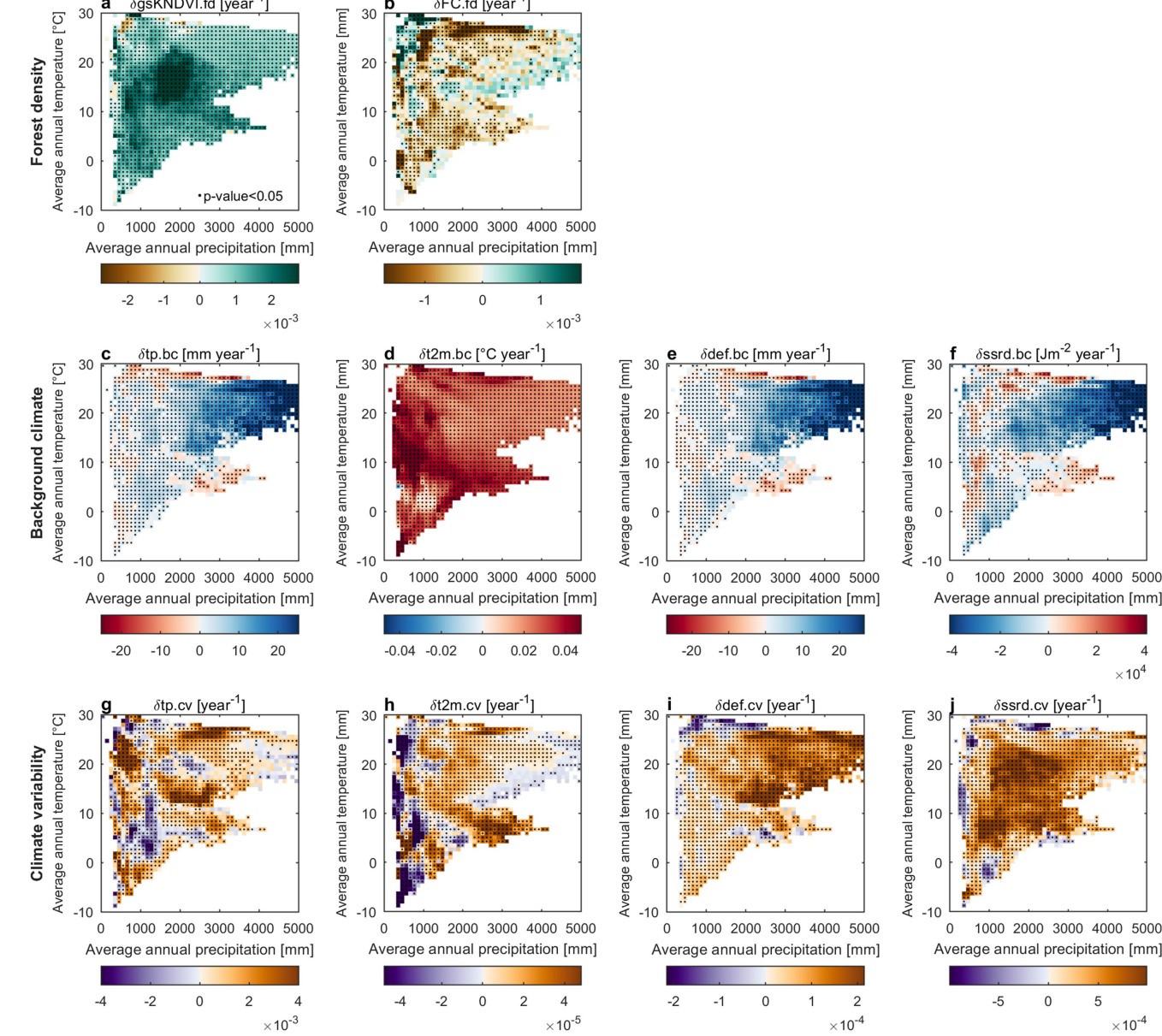

**Extended Data Fig. 7 | Temporal variations in environmental predictors.**
(**a-b**) Temporal trends in environmental predictors of the category 'forest density' computed over a 3-year moving window and binned as a function of climatological precipitation and temperature. Black dots indicate bins with average values that are statistically different from zero (two-sided Student's $t$-test; $P$-value ≤ 0.05). (**c–f**) and (**g–j**) as (**a,b**) but for environmental predictors of the categories 'background climate' and 'climate variability', respectively. Predictor acronyms are reported in Extended Data Table 1.

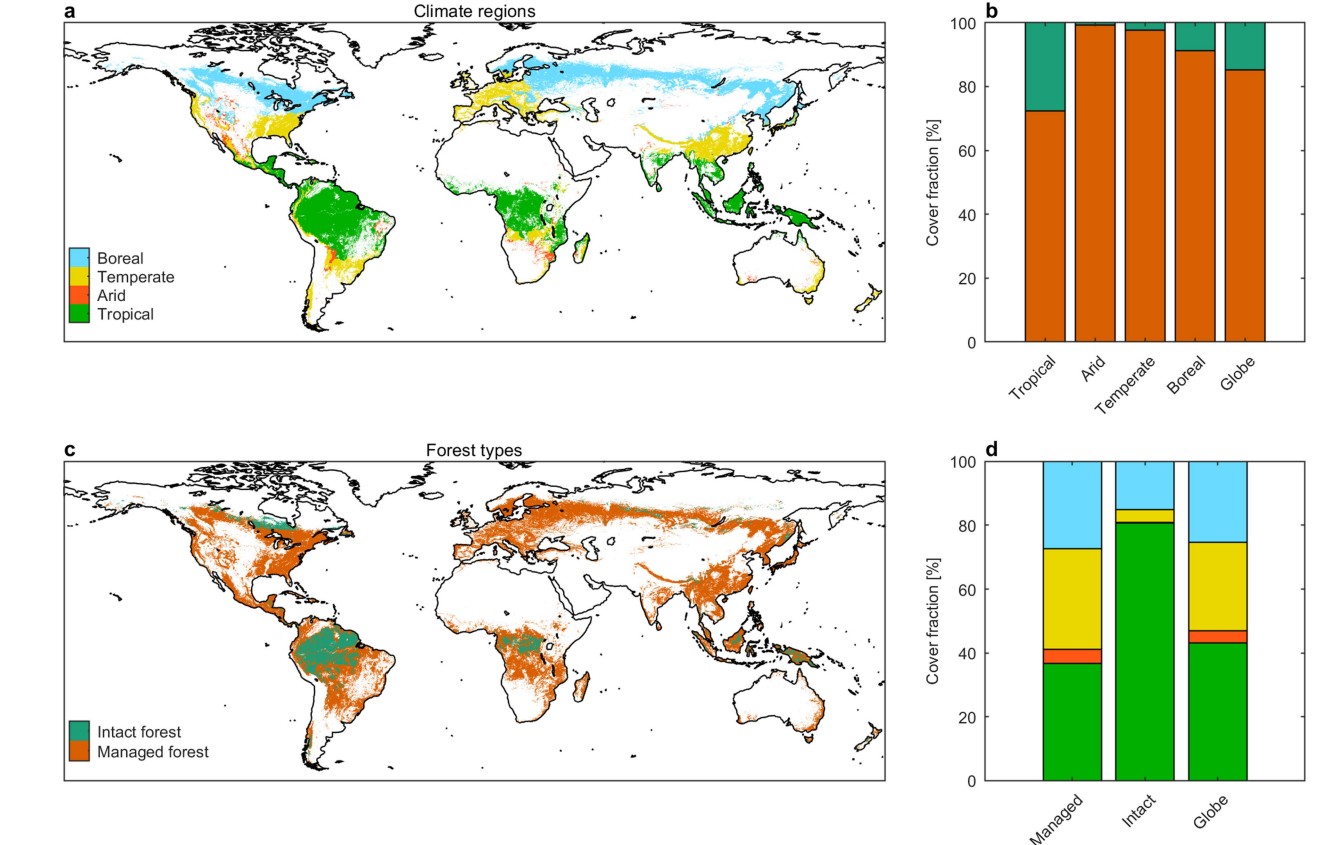

**Extended Data Fig. 8 | Climate and forest domains.** (**a**) Spatial map of climate regions. (**b**) Cover fraction of managed and intact forests for each climate region. (**c**) Spatial map of managed and intact forests. (**d**) Cover fraction of climate regions for each forest domain.

**Extended Data Table 1 | Environmental variables used in the resilience model**

| Variable name | Category | Acronym |
|---|---|---|
| Forest cover | Forest density | FC.fd |
| Growing-season KNDVI | Forest density | gsKNDVI.fd |
| Total precipitation | Background climate | tp.bc |
| Total precipitation | Climate variability | tp.cv |
| Total precipitation | Autocorrelation | tp.ac |
| 2-meter air temperature | Background climate | t2m.bc |
| 2-meter air temperature | Climate variability | t2m.cv |
| 2-meter air temperature | Autocorrelation | t2m.ac |
| Evapotranspiration deficit | Background climate | def.bc |
| Evapotranspiration deficit | Climate variability | def.cv |
| Evapotranspiration deficit | Autocorrelation | def.ac |
| Surface solar radiation downwards | Background climate | ssrd.bc |
| Surface solar radiation downwards | Climate variability | ssrd.cv |
| Surface solar radiation downwards | Autocorrelation | ssrd.ac |

Forest and climate features used as predictors to characterize the response function of forest resilience within a RF predictive model.

**Extended Data Table 2 | Long-term $TAC$ and $\delta TAC$ for different climate regions and forest domains**

| Climate regions | Forest type | Sample size (#pixels) | $TAC$ (2000-2020) | | | $\delta TAC$ | | |
|---|---|---|---|---|---|---|---|---|
| | | | Average | Lower ci | Upper ci | Average | Lower ci | Upper ci |
| **Tropical** | **Managed** | 434485 | 3.34E-01 | 3.33E-01 | 3.35E-01 | 2.00E-03 | 1.98E-03 | 2.03E-03 |
| | **Intact** | 167609 | 6.48E-02 | 6.43E-02 | 6.53E-02 | 1.10E-03 | 1.06E-03 | 1.14E-03 |
| | **Global** | 602094 | 2.58E-01 | 2.57E-01 | 2.58E-01 | 1.63E-03 | 1.61E-03 | 1.66E-03 |
| **Arid** | **Managed** | 61001 | 6.84E-01 | 6.82E-01 | 6.85E-01 | 1.25E-03 | 1.19E-03 | 1.32E-03 |
| | **Intact** | 402 | 6.33E-01 | 6.12E-01 | 6.54E-01 | 1.88E-03 | 9.93E-04 | 2.77E-03 |
| | **Global** | 61403 | 6.83E-01 | 6.82E-01 | 6.84E-01 | 1.26E-03 | 1.19E-03 | 1.33E-03 |
| **Temperate** | **Managed** | 456084 | 4.43E-01 | 4.42E-01 | 4.44E-01 | 1.44E-03 | 1.41E-03 | 1.46E-03 |
| | **Intact** | 9724 | 1.23E-01 | 1.20E-01 | 1.27E-01 | 1.19E-03 | 1.01E-03 | 1.37E-03 |
| | **Global** | 465808 | 4.36E-01 | 4.35E-01 | 4.37E-01 | 1.43E-03 | 1.40E-03 | 1.45E-03 |
| **Boreal** | **Managed** | 541654 | 2.93E-01 | 2.92E-01 | 2.93E-01 | -1.48E-03 | -1.51E-03 | -1.44E-03 |
| | **Intact** | 54164 | 2.31E-01 | 2.30E-01 | 2.32E-01 | -2.14E-03 | -2.25E-03 | -2.03E-03 |
| | **Global** | 595818 | 2.87E-01 | 2.87E-01 | 2.88E-01 | -1.54E-03 | -1.57E-03 | -1.51E-03 |
| **Globe** | **Managed** | 1493224 | 3.73E-01 | 3.72E-01 | 3.73E-01 | 6.54E-04 | 6.38E-04 | 6.71E-04 |
| | **Intact** | 231899 | 9.31E-02 | 9.26E-02 | 9.36E-02 | 7.15E-04 | 6.76E-04 | 7.54E-04 |
| | **Global** | 1725123 | 3.31E-01 | 3.30E-01 | 3.31E-01 | 6.68E-04 | 6.53E-04 | 6.83E-04 |

Values report the average and the corresponding lower and upper 95% confidence intervals (ci). Sample size of each domain is reported in terms of number of 0.05°x0.05° pixels.

# Reporting Summary

## Statistics

For all statistical analyses, confirm that the following items are present in the figure legend, table legend, main text, or Methods section.

| n/a | Confirmed | |
|-----|-----------|---|
| ☒ | ☐ | The exact sample size (*n*) for each experimental group/condition, given as a discrete number and unit of measurement |
| ☒ | ☐ | A statement on whether measurements were taken from distinct samples or whether the same sample was measured repeatedly |
| ☐ | ☒ | The statistical test(s) used AND whether they are one- or two-sided *Only common tests should be described solely by name; describe more complex techniques in the Methods section.* |
| ☐ | ☒ | A description of all covariates tested |
| ☒ | ☐ | A description of any assumptions or corrections, such as tests of normality and adjustment for multiple comparisons |
| ☐ | ☒ | A full description of the statistical parameters including central tendency (e.g. means) or other basic estimates (e.g. regression coefficient) AND variation (e.g. standard deviation) or associated estimates of uncertainty (e.g. confidence intervals) |
| ☐ | ☒ | For null hypothesis testing, the test statistic (e.g. *F*, *t*, *r*) with confidence intervals, effect sizes, degrees of freedom and *P* value noted *Give P values as exact values whenever suitable.* |
| ☒ | ☐ | For Bayesian analysis, information on the choice of priors and Markov chain Monte Carlo settings |
| ☒ | ☐ | For hierarchical and complex designs, identification of the appropriate level for tests and full reporting of outcomes |
| ☒ | ☐ | Estimates of effect sizes (e.g. Cohen's *d*, Pearson's *r*), indicating how they were calculated |

*Our web collection on statistics for biologists contains articles on many of the points above.*

## Software and code

Policy information about availability of computer code

| | |
|---|---|
| Data collection | MATLAB 2017R |
| Data analysis | MATLAB 2017R. The custom MATLAB (R2017b) code written to analyse the data, develop the random forest model and generate figures is available at https://doi.org/10.6084/m9.figshare.19636059.v1. |

For manuscripts utilizing custom algorithms or software that are central to the research but not yet described in published literature, software must be made available to editors and reviewers. We strongly encourage code deposition in a community repository (e.g. GitHub). See the Nature Portfolio guidelines for submitting code & software for further information.

## Data

Policy information about availability of data

All manuscripts must include a data availability statement. This statement should provide the following information, where applicable:

- Accession codes, unique identifiers, or web links for publicly available datasets
- A description of any restrictions on data availability
- For clinical datasets or third party data, please ensure that the statement adheres to our policy

The climate datasets used in this study are publicly available from the ERA5-land reanalysis product (https://cds.climate.copernicus.eu/cdsapp#!/home) and from Köppen-Geiger World map of climate classification (http://koeppen-geiger.vu-wien.ac.at/present.htm). Normalized Difference Vegetation Index (NDVI) data are acquired from the Moderate Resolution Imaging Spectroradiometer (MOD13C1 Version 6, https://lpdaac.usgs.gov/products/mod13c1v006/), land surface phenology data from the Vegetation Index and Phenology (VIP) satellite-based product (https://vip.arizona.edu/) and forest cover data from the European Space Agency's Climate Change Initiative (ESA-CCI, https://www.esa-landcover-cci.org/). Gross Primary Productivity fluxes are available from the FLUXCOM product (http://www.fluxcom.org/) and the spatial delineation of intact forests from the Intact Forest Landscapes dataset (http://intactforests.org/).

# Field-specific reporting

Please select the one below that is the best fit for your research. If you are not sure, read the appropriate sections before making your selection.

☐ Life sciences ☐ Behavioural & social sciences ☒ Ecological, evolutionary & environmental sciences

For a reference copy of the document with all sections, see nature.com/documents/nr-reporting-summary-flat.pdf

# Ecological, evolutionary & environmental sciences study design

All studies must disclose on these points even when the disclosure is negative.

| | |
|---|---|
| Study description | Our contribution provides the first observation-based global-scale assessment of how forest resilience evolved in recent decades in response to global change. For this purpose, we developed a novel methodology that integrates theoretical bases of the resilience of nonlinear dynamical systems approaching a tipping point, satellite-based kernel Normalized Difference Vegetation Index, recently proposed as a strong proxy for ecosystem productivity, and machine learning techniques. |
| Research sample | Spatial and temporal variations in forest resilience were retrieved from MODIS NDVI data, analysed at the pixel scale (0.05°) and aggregated per climate regions (globe, tropical, arid, temperate, boreal) and forest types (managed, intact). Analyses produced at the pixel level were meant to explore the local variations in forest resilience. Analyses conducted separately per climate region and forest type were aimed at characterizing the overall trajectories of forest resilience over large climatically consistent zones and disentangling the human-induced effect on vegetation dynamics (e.g., forest management). The MODIS NDVI product used in this study (MOD13C1 Version 6) being derived from a unique platform and sensor, it is temporally and spatially consistent. Furthermore, the 16-day acquisition time interval and the available temporal coverage (2000-2020) make the MOD13C1 product a suitable dataset to explore spatial and temporal variations in forest resilience. |
| Sampling strategy | When statistics are analyzed at climate region/forest type scale, we included in the sample all pixels that have passed the screening procedure (see section below "Data exclusions"). Sample sizes are typically larger than 50000 pixels (Supplementary Table 2) and therefore are considered fully representative of the climate region/forest type investigated.<br>When results are binned in a 50x50 grid as a function of annual precipitation and temperature (e.g., Fig. 1b), we retained only bins with at least 50 records. Such sampling strategy appears a reasonable compromise to identify major climate features and in parallel to reduce possible noise in bins poorly representative.<br>When differences between managed and intact forests are analyzed, the potential effect of climate background has been removed. To this aim, we compared the climate spaces generated separately for managed and intact forests by extracting only those bins that are covered by both forest classes. The resulting distributions - one for each forest class - have the same sample size and each pair of elements shares the same climate background. This method allows to filter out the potential confounding effect of climate background in the two classes of forests. Similar approach was used to calculate the probability of abrupt decline conditional on negative trend in resilience. The resolution of the climate spaces (50x50 bins) ensures samples of sufficient sizes for statistical analyses. |
| Data collection | All data utilized in this study are acquired from satellite-based datasets and reanalysis products (see section "Data availability"). |
| Timing and spatial scale | We quantify the spatial patterns of forest resilience at the global scale and explore its temporal evolution over the period 2000-2020. |
| Data exclusions | Resilience indicators were derived for forest pixels with less than 50% missing data in the original NDVI data of good and marginal quality and with forest cover greater than 0.05. In order to assess the robustness of our results with respect to the modelling choices described above we performed a series of sensitivity analyses.<br>Sensitivity to the quality flag. The NDVI quality flag (QF) determines the reliability of the original satellite retrievals and therefore affects the robustness of the derived estimates of forest resilience. The quality flags "good" (description: "use with confidence") and "marginal" (description: "useful, but look at other QA information") are typical quality flags utilized for remote sensing applications. In general, estimates based exclusively on the good quality flag are more robust but have lower spatial and temporal coverage compared to those derivable including also data with marginal quality flags. We tested two different quality screening: QF = good and QF = good & marginal.<br>Sensitivity to the percentage of missing data. The percentage of missing data (PMD) allowed at the pixel scale influences the spatial domain of analysis. Pixels with PMD above a fixed threshold are masked out and are excluded from the analyses of forest resilience. Lower values of PMD lead to a smaller spatial domain but characterized by pixels with a higher number of NDVI retrievals, the opposite holds for higher values of PMD. We tested three different PMD thresholds: PMD<20%, PMD<50% and PMD<70%.<br>Sensitivity to the percentage of forest cover. The percentage of forest cover (PFC) allowed at the pixel scale influences the spatial domain of the analysis, similarly to PMD. Pixels with PFC below a fixed threshold are masked out and are excluded from the analyses of forest resilience. Higher values of PFC lead to smaller spatial domain but characterized by pixels more representative of the forest conditions (higher forest extents at pixel level), the opposite holds for lower values of PFC. We tested three different PFC thresholds: PFC>5%, PFC>50% and PFC>90%.<br>Overall assessment. Results of the sensitivity analysis shown in Extended Data Figs. 4-6 and presented in Supplementary Discussion 2 corroborate the robustness of our findings with respect to the modelling choices adopted in our approach (Methods). Therefore, the implemented model setup (QA=good and marginal, PMD<50%, PFC>5%) appears a reasonable compromise to properly capture the spatio-temporal dynamics of forest resilience. |
| Reproducibility | Our data-driven modelling is highly reproducible within the computational accuracy on other computing platforms. |
| Randomization | The Random Forest regression model (RF) we have developed is based on:<br>1) random record selection: each tree is built from a separate random sample of the data using bootstrap sampling. |

2) random predictor selection: in a standard tree, each split is created after examining every predictor and selecting the best split from the number of predictors to sample.
The RF implemented in our study uses 100 regression trees, whose depth and number of predictors to sample at each node were identified using Bayesian optimization.

Blinding n/a

Did the study involve field work? ☐ Yes ☒ No

# Reporting for specific materials, systems and methods

We require information from authors about some types of materials, experimental systems and methods used in many studies. Here, indicate whether each material, system or method listed is relevant to your study. If you are not sure if a list item applies to your research, read the appropriate section before selecting a response.

## Materials & experimental systems

| n/a | Involved in the study |
|-----|----------------------|
| ☒ ☐ | Antibodies |
| ☒ ☐ | Eukaryotic cell lines |
| ☒ ☐ | Palaeontology and archaeology |
| ☒ ☐ | Animals and other organisms |
| ☒ ☐ | Human research participants |
| ☒ ☐ | Clinical data |
| ☒ ☐ | Dual use research of concern |

## Methods

| n/a | Involved in the study |
|-----|----------------------|
| ☒ ☐ | ChIP-seq |
| ☒ ☐ | Flow cytometry |
| ☒ ☐ | MRI-based neuroimaging |

