## [Peer Review File · Nature]

Manuscript Title: Emerging signals of declining forest resilience under climate change

Reviewer Comments & Author Rebuttals

Reviewer Reports on the Initial Version:

Referees' comments:

Referee #1 (Remarks to the Author):

The manuscript by Forzieri et al analyzes vegetation indices from remote sensing data with an autocorrelation based approach to early warning signs. The authors show that there is a concerning decline in stability in important tropical forests.

The paper constitutes a very nice and technical sound piece of large scale data analysis- The results are convincing and deserving of the visibility that a publication in nature would convey. Although autocorrelation-based warning signs have been around for a while their application to large datasets is still a novelty, and the results in this case are important and surprising. In addition to the headline results the manuscript is very rich in minor findings that are also very interesting.

The use of theoretical thought in the manuscript ranging from the main methodology to model considerations and the use of statistics is very nice and greatly adds to the impact. Once published this will be a paper that I would give to students as an example of how theory can inform data analysis.

I only spotted one point that I think deserves a little bit more elaboration is the discussion of distance to the threshold. The argument here is linked to the observation of abrupt declines rather than the autocorrelation itself. There is probably a good reason to do it this way, but the relevant text passages are relatively dense and I felt a more abstract discussion of the reasoning would have helped.

Also as a reader I would have greatly liked to see a timeseries of the original indicators or the TAC somewhere. Although this is not strictly necessary to support the conclusions, an illustrative figure for example using data from a specific grid cell would help intuition a lot.

In summary, this is an interesting and clever paper with an important message. I recommend publication in nature with high priority after some optional minor improvements.

Referee #2 (Remarks to the Author):

The authors performed a global study on forest resilience dynamics since 2000. Resilience metrics are derived from MODIS NDVI time series and are linked (via a RF model) to environmental and climatic drivers. In particular the study focusses on how forest resilience changed over the study period 2000-2020. They show that tropical, arid and temperate forests experience a decline in resilience, linked to the increase in water limitations and climate variability. For boreal forests they find an increasing trend in resilience, attributed to the benefits of climate warming and CO₂ fertilization in cold biomes outweighing the adverse effects of climate change. They further estimate that 22% of intact undisturbed forests have already reached a critical resilience threshold.

Many studies have performed assessments of forest resilience using satellite observations (an obvious reference that is missing from the paper is Gazol et al. 2018, *Global Change Biology*, "Forest Resilience to drought varies across biomes"). Yet, the interesting point this study addresses is the temporal change in forest resilience linked to climate change. Although others have performed similar analysis on a more regional scale e.g. De Keersmaecker et al. (2017), *Remote Sensing*, "Assessing of regional vegetation response to climate anomalies: a case study for Australia using GIMMS NDVI time series", the presented results are relevant for the broader scientific community. As a side comment linked to this. I would suggest to not use too strong wording when stating that 'However, how forest resilience, which is the capacity to withstand and recover from perturbations, is evolving in response to global changes is not yet explored' (L19-21).

The remote sensing analysis of forest resilience seems to be performed according standard procedures published in previous studies. The same holds for the statistical approach to link observed resilience metrics to explanatory variables. Therefore they used a random forest model and as far as I can judge from the text the implementation of the model and statistics was according standard scientific procedures. Nevertheless I have some questions/concerns that needs to be clarified:

- (i) The analysis relies on the kNDVI (based on NDVI time series). It is common knowledge that NDVI tend to saturate in forests. When calculating anomalies from these time series you end up with very noisy signals. Is the kNDVI approach dealing with all of this? EVI is often used to address this issue.
- (ii) You perform a global analysis. However, can you be assured that your kNDVI signals are consistent across climatic regions and biomes? This again links to my former comment. I'm a bit worried about the noise sensitivity of your analysis. Are you sure that the differences you find between biomes are not the results of higher or lower noise levels in the anomaly times series or TAC time series rather than an actual difference in ecosystem functioning?
- (iii) Please clarify: "Long-term TAC can only partially characterize forest resilience, since this ecosystem property may evolve in response to changing environmental conditions" (L102-103).
- (iv) You analyzed the temporal evolution of TAC based on a 3-year rolling window. I'm not sure where the 3 year window is coming from? Is this the optimal time window to calculate temporal dynamics in TAC? Is your TAC metrics and the delta TAC still valid at such short time windows?
- (v) In your selection of MODIS images you also include data with marginal overall quality. Why do you do this? Why not only include data of good quality? What is the impact of including these lower quality observations?

- (vi) Please clarify: “Missing data, due for instance to snow cover, have been filled by local climatological values.” (L287-288)
- (vii) It is not entirely clear to me how you statistically deal with spatial autocorrelation (in your TAC). Somehow you need to deal with that either within the RF model statistics or through a smart sampling design (selection of a subset of all your pixels).
- (viii) Linked to this: make sure that throughout the manuscript it is always clear to the reader when you are referring to spatial and when to temporal autocorrelation. Now it was not always clear to me.
- (ix) Please include some lines of text about the outcome/result of your sensitivity analysis. Now you just refer to a figure but you’re not informing the reader about the outcome of the sensitivity analysis. E.g. what’s the impact of your choice of accepting up to 50% NA’s in your time series. Not better to only allow 10%?

Referee #3 (Remarks to the Author):

Review comments

The manuscript by Forzieri et al. quantifies global changes in vegetation indices and derived indicators and uses these to draw a wide range of inferences about changes in global forest resilience.

While I agree with the fundamental statements of the authors that forests are at increasing risks from climate-change induced disturbances and that this needs to be taken into consideration when designing land-based mitigation plans, I am not yet convinced that the data shown by the authors support the conclusions centered on decreasing resilience.

The authors have conducted a very impressive amount of technical analyses but their manuscript overwhelms the readers with mountains of data and with inadequate and insufficient explanations of the ecological foundations on which the authors base the interpretation of their data.

It is essential that the authors more clearly explain why changes in a remotely sensed vegetation index over a 21 year period are indicators of forest resilience, defined by the authors as “capacity to withstand and recover from perturbations”. The explanations of the methods provide insufficient information to the readers about the various indices and the justification for the ecological interpretations assigned by the authors. For example, the authors do not know when disturbances occurred prior to their observation period (pre-2000). The authors therefore have to limit their analysis of recovery to forests that were disturbed since 2000. What proportion of pixels contributes to the analysis of recovery (and changes in recovery rates)? In boreal forests, a 21-year observation period is simply insufficient to draw any inferences about CHANGES in recovery rates. It is likely not even sufficient to estimate recovery rates. If pre-2000 disturbances and subsequent recovery estimates are included in the analyses then the authors need to state this.

The authors must provide much clearer descriptions of how the remote sensing indicators relate to ecological processes. The statement that “Specifically, we retrieved the 1-lag temporal autocorrelation (TAC) as a CSD indicator related to resilience^{3–5} from the satellite-based kernel Normalized Difference Vegetation Index (kNDVI) derived for the 2000-2020 period at the global scale at 0.05° spatial resolution from the MODIS sensor” is not transparent. References 3 – 5 explain

resilience, but they do not justify why the TAC is an indicator of resilience.

The authors state that “High TAC values (e.g., arid forests) implies [sic] low recovery rates and thus low resilience”. Ecosystems where environmental conditions (water, temperature) limit growth rates and therefore limit rates of recovery are not inherently less resilient to perturbations. Boreal forests have been subject to wildfire disturbances for millennia and simply because they grow more slowly than tropical forests does not imply that they are less resilient. In fact, most boreal forest ecosystems have evolved with fires and have developed recovery strategies adapted to boreal conditions, such as serotinous cones in black spruce.

The authors of this global study state that “High TAC implies low recovery rates and thus low resilience, the opposite holds for low TAC values.” Recovery rates are highly specific to ecosystem types and productivity. These will be much faster in tropical or subtropical regions than in temperate regions and forests in all three regions will recover faster than forests in boreal regions. Without some reasonable baseline information, a 21-year observation period is insufficient to draw any conclusions about CHANGES in recovery rates in boreal forests.

The authors write “The results demonstrate that intact forests have considerably lower long-term TAC (i.e. higher average forest resilience) than managed forests (0.13 and 0.21, respectively, Fig. 3a). This finding reinforces the expectation that intact forests have a higher capacity to withstand external perturbations thanks to their typically higher structural complexity and species richness^{23,24}”. The sweeping generalizations and speculations of the impacts of forest management on resilience also require further justifications. In particular because the map of intact vs managed forest is not credible (supplementary material Fig 5). The authors explain that they use a map source that identifies intact forests. “The remaining forests pixels - not labelled as intact - were considered as managed forests (Supplementary Fig. 5).” This approach yields some absurd results: for example, large regions of northern Alaska, northern Canada (near the northern tree line) and vast regions of northern Russian boreal forests are shown as managed. This is simply wrong. Consequently, I have no confidence in any of the claims made about resilience of managed vs intact forests. The authors either have to use a more credible definition of managed forests (and could consider those used for UNFCCC reporting – Ogle et al. 2018 <https://doi.org/10.1186/s13021-018-0095-3>) or they have to remove this very speculative part of their analysis prior to publication.

Figure 3, which contains 6 panels, is intended to specify the effects of forest management on forest resilience. Yet the definition of forest management is based on Figure 5 of the supplementary material, which is fundamentally wrong.

Moreover, evidence from 30m resolution time series of forest recovery in boreal forests shows that recovery following harvest (i.e., management) is more rapid than recovery following wildfire (White et al. 2017 <http://dx.doi.org/10.1016/j.rse.2017.03.035>). How do the authors reconcile this observation with their claim that resilience in managed boreal forests is lower than in intact forests? Page 5: “Results show that, at the global level, intact forests have a probability of AD conditional on high δ TAC greater than 0.5 (Fig. 4a). This signal is statistically significant and increases with the severity of AD suggesting that the progressive deterioration of ecosystem states, as tracked by the decline of resilience, has likely contributed to the upsurge of negative anomalies. The emerging relation is mainly driven by boreal forests, particularly those in central Russia, where there is emerging a localized decline in forest resilience (Fig. 1c).” The drivers of the most abrupt declines (AD) in boreal forests of Canada and Russia (together by far the largest proportion of global boreal forests) will be wildfires and harvest. Since harvest always targets mature forests are you suggesting that mature forests that are approaching harvest age show signs of declining resilience? But, in your

analyses you are not differentiating the drivers of AD. Wildfire, in contrast to harvest, can affect stands of all ages. In boreal forests most of the area burned is ignited by lightning strikes whose location is not influenced by the “resilience” of forests. You may be observing some statistical correlations here, but I am not at all convinced that your signal has any ecological meaning or is an indicator of decline of forest resilience.

On page 6 the authors write that: “These hypotheses are also consistent with the dominant climate drivers of δTAC in boreal and tropical regions (background climate and climate variability, respectively, Fig. 2b,c) and further supported by multiple independent evidences (e.g., refs(30,43–45)). Do the authors have any independent references for boreal forests? References 30, 43-45 are all studies from the tropics.

Page 6: “On the contrary, benefits induced by climate warming and CO₂ fertilization have outweighed such negative effects in boreal regions, ultimately leading to an improvement of forest resilience.” How do the authors reconcile large increases in the areas affected by wildfires and insects in global boreal forests with their claim that forest resilience has increase over the observation period? This contradiction has to be addressed by the authors.

Concluding paragraph: “Therefore, it is becoming urgent to account for these trends in the design of effective forest-based mitigation strategies to avoid future negative surprises triggered by the increasing vulnerability of carbon stocks.” I completely agree with the statement. Forest mitigation strategies that do not consider vulnerability of carbon stocks, or worse, that lead to increased vulnerability of C stocks are counterproductive to effective climate mitigation. However, the authors have not yet convinced me that their measure of forest resilience is a suitable to detect the vulnerability of carbon stocks.

Three additional comments: (1) it is frustrating that authors submit a manuscript without line numbers. How are reviewers expected to comment on statements when all we can reference is a page number?

(2) There are a number of typos, grammatical errors and incorrect uses of words, which I would normally point out but without line numbers this is not worth my time.

(3) Figure 2b – the most common color-blindness in the human population is red-green. The authors could greatly improve this graph by replacing one of the red or green lines with a blue one and further modified the line types or added distinct markers so that readers do not have to rely on color alone. See also Nature 598, 224-225 (2021). doi: <https://doi.org/10.1038/d41586-021-02696-z>.

In summary, I cannot support publication of the manuscript in the current form in Nature and recommend revisions and resubmission of the manuscript to a journal that specialises in remote-sensing analyses.

Author Rebuttals to Initial Comments:

First of all, we would like to thank the three referees for their insightful and constructive comments. In our revision of the manuscript, we tried to address all their comments and suggestions in order to improve the robustness of the analysis and the clarity of the interpretation. In the following text, we respond to each reviewer's comment by referring to line numbers of the revised tracked version, when not differently indicated. References cited along these responses are reported at the bottom of the document.

Referee #1

The manuscript by Forzieri et al analyzes vegetation indices from remote sensing data with an autocorrelation based approach to early warning signs. The authors show that there is a concerning decline in stability in important tropical forests. The paper constitutes a very nice and technical sound piece of large scale data analysis- The results are convincing and deserving of the visibility that a publication in nature would convey. Although autocorrelation-based warning signs have been around for a while their application to large datasets is still a novelty, and the results in this case are important and surprising. In addition to the headline results the manuscript is very rich in minor findings that are also very interesting.

The use of theoretical thought in the manuscript ranging from the main methodology to model considerations and the use of statistics is very nice and greatly adds to the impact. Once published this will be a paper that I would give to students as an example of how theory can inform data analysis.

I only spotted one point that I think deserves a little bit more elaboration is the discussion of distance to the threshold. The argument here is linked to the observation of abrupt declines rather than the autocorrelation itself. There is probably a good reason to do it this way, but the relevant text passages are relatively dense and I felt a more abstract discussion of the reasoning would have helped.

Action taken. According to the reviewer's comment, we have clarified the analysis of the thresholds and tipping points in the main text (see lines 209-210 and lines 220-222). Furthermore, we have added two dedicated sections in the Supplementary Material to describe the theoretical framework adopted in our study (Supplementary Methods 1 "Thresholds and tipping points" and Supplementary Methods 2 "Relationship between resilience, critical slowing down and early warning signals").

Also as a reader I would have greatly liked to see a timeseries of the original indicators or the TAC somewhere. Although this is not strictly necessary to support the conclusions, an illustrative figure for example using data from a specific grid cell would help intuition alot.

Action taken. Supplementary Figure 5 shows the time series of TAC aggregated for climate regions and for the whole globe. Furthermore, we have added two illustrative figures that describe the theoretical framework of critical slowing down indicators (Supplementary Figs. 1 and 2) to complement the new methodological sections (Supplementary Methods 1 and 2).

In summary, this is an interesting and clever paper with an important message. I recommend publication in nature with high priority after some optional minor improvements.

We thank the reviewer for her/his positive comments.

Referee #2:

The authors performed a global study on forest resilience dynamics since 2000. Resilience metrics are derived from MODIS NDVI time series and are linked (via a RF model) to environmental and climatic drivers. In particular the study focusses on how forest resilience changed over the study period 2000-2020. They show that tropical, arid and temperate forests experience a decline in resilience, linked to the increase in water limitations and climate variability. For boreal forests they find an increasing trend in resilience, attributed to the benefits of climate warming and CO2 fertilization in cold biomes outweighing the adverse effects of climate change. They further estimate that 22% of intact undisturbed forests have already reached a critical resilience threshold.

Many studies have performed assessments of forest resilience using satellite observations (an obvious reference that is missing from the paper is Gazol et al. 2018, Global Change Biology, "Forest Resilience to drought varies across biomes"). Yet, the interesting point this study addresses is the temporal change in forest resilience linked to climate change. Although others have performed similar analysis on a more regional scale e.g. De Keersmaecker et al. (2017), Remote Sensing, "Assessing of regional vegetation response to climate anomalies: a case study for Australia using GIMMS NDVI time series", the presented results are relevant for the broader scientific community.

As a side comment linked to this. I would suggest to not use too strong wording when stating that 'However, how forest resilience, which is the capacity to withstand and recover from perturbations, is evolving in response to global changes is not yet explored' (L19-21).

Action taken. According to the reviewer's suggestion, we have rephrased the text as: "However, little is known about how forest resilience, which is the capacity to withstand and recover from perturbations, is evolving in response to global changes.". Furthermore, we have added the two suggested references when presenting the background of the study.

The remote sensing analysis of forest resilience seems to be performed according standard procedures published in previous studies. The same holds for the statistical approach to link observed resilience metrics to explanatory variables. Therefore they used a random forest model and as far as I can judge from the text the implementation of the model and statistics was according standard scientific procedures. Nevertheless I have some questions/concerns that needs to be clarified:

We have answered to each specific question/concern in the following lines.

(i) The analysis relies on the kNDVI (based on NDVI time series). It is common knowledge that NDVI tend to saturate in forests. When calculating anomalies from these time series you end up with very noisy signals. Is the kNDVI approach dealing with all of this? EVI is often used to address this issue.

The relationship between NDVI and green biomass is indeed highly nonlinear and tends to saturate for high NDVI values. Some indices such as the enhanced vegetation index (EVI) (ref. ⁽¹⁾) have tried to compensate for this using information from other bands, but the saturation problem remains. The saturation at high values of the signal hampers the calculation of the temporal autocorrelation of the signal, since for these indexes the amplitude of temporal variations is affected by the intensity.

kNDVI has been recently proposed to address the above-mentioned saturation problem of classical indexes like EVI and NDVI, together with other limitations of existing remote sensing vegetation indices, by exploiting all higher-order relations between the spectral channels involved². This aspect is extensively discussed in the paper that presented this new metric².

Overall, kNDVI has been documented to be more resistant to saturation, bias, and complex phenological cycles and shows enhanced robustness to noise and stability across spatial and temporal scales compared to alternative products (e.g., NDVI, NIRv). In addition, kNDVI proved to be much closer and linearly correlated to GPP than other spectral indexes of primary productivity (e.g. experimental evidences have been produced both against eddy covariance estimates of GPP at Fluxnet sites and against satellite SIF retrievals). Thanks to the linearity between kNDVI and primary productivity, this metric is clearly more suited than classical spectral indexes for the calculation of TAC. For this reason, kNDVI was selected and used in the analysis.

Action taken. We have clarified the specific properties and advantages of kNDVI as a metric of ecosystem primary productivity for the calculation of resilience in the Method section (see lines 279-285) and briefly reported in the main text as well (lines 79-81 and lines 112-116).

(ii) You perform a global analysis. However, can you be assured that your kNDVI signals are consistent across climatic regions and biomes? This again links to my former comment. I'm a bit worried about the noise sensitivity of your analysis. Are you sure that the differences you find between biomes are not the results of higher or lower noise levels in the anomaly times series or TAC time series rather than an actual difference in ecosystem functioning?

This is an important point that we carefully evaluated when designing the analysis. Spectral signals from satellite platforms are indeed prone to noise and potential biases that may affect the analysis of the temporal variations in the signal when data are derived from multiple satellite platforms. **Ultimately we selected kNDVI from MODIS because 1) it proved to be robust across large spatial scales against different datasets (see ref.⁽²⁾ for a detailed evaluation of the metric at global scale against SIF measurements and at Fluxnet sites) 2) it shows consistently better correlations with other productivity metrics than other spectral indexes, and 3) it is based on a single satellite platform that has been operated for two decades.**

In addition, **the main finding of our study related to the temporal variability of the signal (decline in TAC) and therefore the results are not affected by the spatial variability.** In fact, the entire time series used in the analysis is coming from a single sensor and technology (MODIS) and therefore temporal variation of autocorrelations are indeed to be attributed to a changes in vegetation properties. Thanks to the specific properties of the index and to the stability of the satellite platform, we are confident that the signals we retrieved are not an artefact of noise in the signals.

Action taken. We have clarified in the text the specific properties of kNDVI from MODIS and explained why it is a robust productivity metric that fulfills the specific requirements of the analysis in the Method section (see lines 279-285) and briefly in the main text as well (lines 79-81 and lines 112-116).

(iii) Please clarify: “Long-term TAC can only partially characterize forest resilience, since this ecosystem property may evolve in response to changing environmental conditions” (L102-103).

Action taken. We have substantially revised this paragraph and clarified the text (see lines 82-93 and lines 305-310). Furthermore, to better focus on the main findings of our study (temporal changes in forest resilience) we have moved to supplementary material most of results related to the long-term TAC (see Supplementary Discussion 1 and Supplementary Fig. 3 and 4).

(iv) You analyzed the temporal evolution of TAC based on a 3-year rolling window. I’m not sure where the 3 year window is coming from? Is this the optimal time window to calculate temporal dynamics in TAC? Is your TAC metrics and the delta TAC still valid at such short time windows?

We performed a set of preliminary experiments to explore the temporal evolution of TAC using a temporal rolling window ranging from 1 to 3 years. We adopted the 3-year window for the analyses shown in the manuscript as a reasonable compromise between obtaining a robust estimate of the short-term TAC and exploring his variation over the 21-year observational period. Indeed, shorter temporal windows tend to provide a less robust estimate of TAC, whereas longer temporal windows reduce the length of the time series to be analyzed, therefore limiting the possibility to assess trends. **It is important to stress that by using different rolling windows the main findings and conclusions of the analysis do not change.**

Action taken. We have clarified our approach and have performed a set of additional experiments to test the sensitivity of our results on the length of the rolling windows (lines 395-408). Results of these analyses were discussed in a new dedicated section (Supplementary Discussion 2) and visualized in a new figure (Supplementary Figure 7).

(v) In your selection of MODIS images you also include data with marginal overall quality. Why do you do this? Why not only include data of good quality? What is the impact of including these lower quality observations?

We opted to include pixels with good and marginal overall NDVI quality to build statistical inferences on larger samples without the detrimental effect of potential noise originating from pixels with low NDVI quality data. **We tested this against a more conservative approach and concluded that the exclusion of pixels with marginal overall NDVI quality would have not affected our results and conclusions.**

Action taken. We have clarified this in the text (see lines 395-408) and added a new dedicated section in Supplementary Material to discuss the sensitivity of our results on the NDVI quality (Supplementary Discussion 2). Results are shown in the new Supplementary Figures 5 and 6.

(vi) Please clarify: “Missing data, due for instance to snow cover, have been filled by local climatological values.” (L287-288)

Action taken. Missing data, due for instance due to snow cover affecting the retrieval of reflectance properties, have been filled before analysis by climatological kNDVI values. According to the reviewer's comment, we have clarified this in the revised version (see lines 289-291). We have also tested the sensitivity of our results on the gap filling procedure adopted. Results of these additional experiments confirm the validity our approach and the robustness of our findings. Results of the sensitivity analyses are discussed in a new dedicated section (Supplementary Discussion 2) and shown in Supplementary Figures 5 and 6.

(vii) It is not entirely clear to me how you statistically deal with spatial autocorrelation (in your TAC). Somehow you need to deal with that either within the RF model statistics or through a smart sampling design (selection of a subset of all your pixels).

The autocorrelation function is always computed over time (not in space). Temporal trends in autocorrelation are used as an indicator of forest resilience in our study. For this reason the spatial dimension of the correlation does not affect the results and conclusions. On this subject, see also our response to your next comment.

With regard to the potential spatial correlation of MODIS reflectance deriving from the multi-angular view of the sensor, we argue that they are rather limited at the coarse spatio-temporal resolution of the analysis (about 5km, 16 days composite), given that the original retrievals are performed at the resolution of 500m, daily. In addition, the main objective of the paper is to assess temporal variation in the signal at coarse resolution pixels. The temporal variation of the retrieved reflectances are not affected by spatial autocorrelation in MODIS retrievals.

Some technical specification on the derivation of the metrics follows. TAC is computed on kNDVI anomalies by utilizing MODIS-derived NDVI provided at 16-day temporal resolution and 0.05° spatial resolution (MOD13C1). The 16-day composite NDVI is generated using the two 8-day composite surface reflectance granules in the 16-day period originally acquired at 500-meter spatial resolution (MOD09A1).

Action taken. We have clarified this in the revised version (see for instance lines 76, 82, 92, 305, 353, 354).

(viii) Linked to this: make sure that throughout the manuscript it is always clear to the reader when you are referring to spatial and when to temporal autocorrelation. Now it was not always clear to me.

Please note that in the study we did not assess spatial autocorrelation because it was not relevant for the objective of the analysis. **The autocorrelation function is always computed over time** and used as indicator of forest resilience. In line with the literature on the topic (e.g., refs. (3-14)), we refer to forest resilience as the capacity of the system to remain in its current state despite external perturbations (ecological resilience, ref.(15)). Forest ecosystems are continuously subject to external perturbations (e.g., inter-annual variation of climate drivers), and the capacity of the system to respond to such pressures defines its resilience, which we quantify as changes in **temporal autocorrelation** following well consolidated approaches (e.g., refs. (8,9,16)).

We initially explored the average recovery rates by computing the temporal autocorrelation function at the pixel level from the whole kNDVI time series 2000-2020 (long-term *TAC*). Output of this analysis provides insights on the spatial patterns of slowness and its dependence on environmental factors. We then quantify the temporal variations in forest resilience to explore how forest response is evolving to climate changes by quantifying the changes in *TAC* over two subsequent and independent temporal windows (2000-2010 and 2011-2020) and over a 2-year lagged moving window. In all the afore-mentioned analyses, we focused on the temporal autocorrelation, not on spatial autocorrelation.

Action taken. According to the reviewer's comment, we have clarified this in the revised version (see for instance lines 76, 82, 92, 305, 353, 354). Furthermore, we have added two dedicated sections in Supplementary Material (Supplementary Methods 1-2 and Supplementary Figures 1 and 2) to clarify the theoretical framework of critical slowing down indicators and the potential of temporal autocorrelation (*TAC*) as early warning signal of critical transitions.

(ix) Please include some lines of text about the outcome/result of your sensitivity analysis. Now you just refer to a figure but you're not informing the reader about the outcome of the sensitivity analysis. E.g. what's the impact of your choice of accepting up to 50% NA's in your time series. Not better to only allow 10%?

Action taken. According to the reviewer's comment, we have discussed the results of the sensitivity analysis. To this aim, we have added a dedicated section in the Supplementary Material (Supplementary Discussion 2 Results of the sensitivity analysis). Based on the reviewer's suggestions (see previous reviewer's comments) we have extended this analysis by testing the robustness of our results with respect to the NDVI quality, the gap filling procedure and the length of the temporal windows. Supplementary Figure 5 has been modified and two additional figures (Supplementary Figures 6 and 7) have been produced to include the new experiments.

Referee #3:

The manuscript by Forzieri et al. quantifies global changes in vegetation indices and derived indicators and uses these to draw a wide range of inferences about changes in global forest resilience. While I agree with the fundamental statements of the authors that forests are at increasing risks from climate-change induced disturbances and that this needs to be taken into consideration when designing land-based mitigation plans, I am not yet convinced that the data shown by the authors support the conclusions centered on decreasing resilience. The authors have conducted a very impressive amount of technical analyses but their manuscript overwhelms the readers with mountains of data and with inadequate and insufficient explanations of the ecological foundations on which the authors base the interpretation of their data.

Our main goal of assessing the temporal changes in forest resilience from satellite retrievals at the global scale, and quantifying its key drivers, requires the use of **big-data analytics, an emerging research field to exploit the expanding availability of Earth observations at high spatial and temporal resolution**. Our approach is based on such methods, building on the computation of resilience indicators from satellite retrievals of proxies of primary productivity. The analysis further relies on a machine-learning framework to disentangle the changes in resilience from the direct climate effects on vegetation. We stress that the large amount of data and numerical analysis presented in the paper is typical of this type of study and is key to exploit the potential of big-data in environmental assessment.

We stress that the ecological foundations on which our interpretations were developed are grounded on solid theoretical and empirical studies on the phenomenon known in dynamical systems theory as “critical slowing down”, which has been demonstrated to be a powerful indicator of resilience and widely adopted in previous research on forest ecosystems and other contexts as well (e.g., climate, finance) (e.g., refs. (3–14)).

In the following answers, we will provide additional clarifications to demonstrate that our methodology is appropriate and the results are robust.

It is essential that the authors more clearly explain why changes in a remotely sensed vegetation index over a 21 year period are indicators of forest resilience, defined by the authors as “capacity to withstand and recover from perturbations”. The explanations of the methods provide insufficient information to the readers about the various indices and the justification for the ecological interpretations assigned by the authors.

For example, the authors do not know when disturbances occurred prior to their observation period (pre-2000). The authors therefore have to limit their analysis of recovery to forests that were disturbed since 2000. What proportion of pixels contributes to the analysis of recovery (and changes in recovery rates)? In boreal forests, a 21-year observation period is simply insufficient to draw any inferences about CHANGES in recovery rates. It is likely not even sufficient to estimate recovery rates. If pre-2000 disturbances and subsequent recovery estimates are included in the analyses then the authors need to state this.

We believe that the reviewers’ conclusion originates from a misunderstanding about the concept of resilience used in the paper.

The reviewer considers resilience as the capacity of an ecosystem to recover after the occurrence of a natural or anthropogenic disturbance event that leads the system in a

different state (e.g., wildfires, insect and harvest). We completely agree that a 21-year observation period would not be sufficient to quantify changes in post-disturbance recovery, not only in boreal forests, as stressed by the reviewer, but probably in any forest of the globe (see for instance ref. (17)). However, this definition of resilience is not the only one used in the literature¹⁸. In fact, **resilience is also defined as the capacity of the system to remain in its current state despite external perturbations (ecological resilience, ref. (15)) that do not lead to a shift in the ecosystem state as disturbances do.** More importantly, measuring resilience under this second definition does not require the occurrence of a single strong external perturbation, and it is mathematically equivalent to measuring short-term responses to weak continuous external perturbations¹⁹.

In line with the literature on the topic (e.g., refs. (3–14)), we use the latter concept of resilience in our study. Forest ecosystems are continuously subject to external perturbations (e.g., inter-annual variation of climate drivers), and the capacity of the system to respond to such pressures defines its resilience, which we quantify as changes in temporal autocorrelation following well consolidated approaches (e.g., refs. (8,9,16)). Therefore, resilience in our work is a broader property of forests to withstand perturbations and avoid state shifts and not the capacity to recover from a state-shift. **To this aim, a 21-year observation period is therefore a reasonably long temporal coverage to explore changes in forest resilience.**

Action taken. We have clarified the resilience concepts and indicators we used in the main text (see lines 55-62). Furthermore, to clarify the theoretical framework adopted in our study, we have added three dedicated methodological sections in Supplementary Material (Supplementary Methods 1-3 “Thresholds and tipping points”, “Relationship between resilience, critical slowing down and early warning signals” and “Forest resilience and disturbance regimes”) and two illustrative figures (Supplementary Figures 1 and 2).

The authors must provide much clearer descriptions of how the remote sensing indicators relate to ecological processes. The statement that “Specifically, we retrieved the 1-lag temporal autocorrelation (TAC) as a CSD indicator related to resilience^{3–5} from the satellite-based kernel Normalized Difference Vegetation Index (kNDVI) derived for the 2000-2020 period at the global scale at 0.05° spatial resolution from the MODIS sensor” is not transparent. References 3 – 5 explain resilience, but they do not justify why the TAC is an indicator of resilience.

We do not agree with the reviewer. In particular:

- We do not understand how the reported statement (“Specifically, we retrieved [...] from the MODIS sensor”) can be considered not transparent. All the relevant information are reported in the statement. Additional details are provided in Supplementary Material and in the mentioned references.
- Regarding the mentioned references 3-5. The reviewer argues that they only explain resilience but do not justify why TAC is an indicator of resilience. This is not true. The referred articles provide the context where TAC has been introduced as an indicator of resilience. See for instance ref. (4) box 1 e ref. (5) figure 1 and related text. The mentioned references explain the theoretical context on which our approach is built.

Action taken. We have clarified the concept of ecological resilience by improving the main text and adding three dedicated sections in Supplementary Material (Supplementary Methods 1-3) about the theoretical framework adopted in our study and included two new illustrative figures (Supplementary Figures 1 and 2).

The 3-5 references in the above-mentioned text (consistent with the initially submitted version of the manuscript) correspond to the following reference numbers reported at the bottom of this document: (3) → ⁽⁸⁾ | (4) → ⁽⁹⁾ | (5) → ⁽³⁾

The authors state that “High TAC values (e.g., arid forests) implies [sic] low recovery rates and thus low resilience”. Ecosystems where environmental conditions (water, temperature) limit growth rates and therefore limit rates of recovery are not inherently less resilient to perturbations. Boreal forests have been subject to wildfire disturbances for millennia and simply because they grow more slowly than tropical forests does not imply that they are less resilient. In fact, most boreal forest ecosystems have evolved with fires and have developed recovery strategies adapted to boreal conditions, such as serotinous cones in black spruce. The authors of this global study state that “High TAC implies low recovery rates and thus low resilience, the opposite holds for low TAC values.” Recovery rates are highly specific to ecosystem types and productivity. These will be much faster in tropical or subtropical regions than in temperate regions and forests in all three regions will recover faster than forests in boreal regions

We agree with the reviewer. Indeed, the long-term *TAC* quantifies the average forest recovery rates and should not be viewed as an absolute metric of forest resilience. The signal, by integrating the interplay between forest and climate, reflects the slowness of variation in forest productivity indexes, as mediated by environmental factors that affect plant growth rates and their capacity to recover from perturbations. **Admittedly, we have over-interpreted the long-term *TAC* in our initial version of the manuscript.** Indeed, low recovery rates do not imply that the system will reach a tipping point earlier compared to a forest located in a different area with high recovery rates. Such mechanisms depend on specific ecological tolerance limits and adaptation properties as extensively discussed in the main text (see lines 204-217) and shown in Fig. 3c. **These results are fully in line with the reviewer’s expectations.**

However, we point out that discussion and interpretation of the sensitivity of the long-term *TAC* on a suite of environmental factors, including those variables that contribute to growth rate, are derived from partial dependence plots (PDPs) generated from the machine learning algorithm RF. This model-agnostic method enabled us to isolate the marginal effect of each environmental predictor and therefore to understand how each driver affects recovery rates. **Discussion based on PDPs are therefore correct.**

Furthermore, we emphasize that our study focuses on the analysis of the changes in forest resilience over time. To this aim, we analysed the temporal evolution of *TAC* computed on kNDVI with 3-year rolling windows over the observational period. In such analysis, the observed trends in forest resilience derived at the pixel-level are exclusively based on how the system is evolving locally. **The comparison of different areas characterized by different trends in *TAC* is therefore not affected by the local absolute level of *TAC* and does not affect the conclusions of our work.**

With regard to the example of adaptation strategy mentioned by the reviewer (serotinous cones in black spruce), we recognize the relevance of long-term forest ecosystem strategies to influence their evolutionary processes in response to recurrent disturbances. However, **our goals and methods are not aimed at capturing such long-term evolutionary processes, which cannot be observed over a 21-year observation period. In our study, we focus on the short-term responses of forests to perturbations that can inform on the ongoing trend in resilience to climate change.**

Action taken. According to the reviewer's comment, we have revised the interpretation of the long-term *TAC* (see lines 88-91) and the description of the corresponding methods ("Spatial patterns of slowness and its dependence on environmental factors", lines 304-314). Part of the contents related to the long-term *TAC* metric (interpretation of PDPs) have been moved to Supplementary Discussion 1 and Supplementary Fig. 3 to fit the length requirements of the journal and to better focus the narrative on the main findings, which relate to the changes in forest resilience over time. A paragraph has been added in the method section ("Critical slowing down indicators") to highlight the relevance of the long-term forest ecosystem processes (lines 391-393).

Without some reasonable baseline information, a 21-year observation period is insufficient to draw any conclusions about CHANGES in recovery rates in boreal forests.

This comment is linked to the misunderstanding of reviewer about the resilience concept adopted in our work. We have already discussed this criticism in previous answers.

The authors write "The results demonstrate that intact forests have considerably lower long-term TAC (i.e. higher average forest resilience) than managed forests (0.13 and 0.21, respectively, Fig. 3a). This finding reinforces the expectation that intact forests have a higher capacity to withstand external perturbations thanks to their typically higher structural complexity and species richness^{23,24}". The sweeping generalizations and speculations of the impacts of forest management on resilience also require further justifications. In particular because the map of intact vs managed forest is not credible (supplementary material Fig 5). The authors explain that they use a map source that identifies intact forests. "The remaining forests pixels - not labelled as intact - were considered as managed forests (Supplementary Fig. 5)." This approach yields some absurd results: for example, large regions of northern Alaska, northern Canada (near the northern tree line) and vast regions of northern Russian boreal forests are shown as managed. This is simply wrong. Consequently, I have no confidence in any of the claims made about resilience of managed vs intact forests. The authors either have to use a more credible definition of managed forests (and could consider those used for UNFCCC reporting – Ogle et al. 2018 <https://doi.org/10.1186/s13021-018-0095-3>) or they have to remove this very speculative part of their analysis prior to publication. Figure 3, which contains 6 panels, is intended to specify the effects of forest management on forest resilience. Yet the definition of forest management is based on Figure 5 of the supplementary material, which is fundamentally wrong.

Concerning the map of managed forest at higher latitude, we realized an error in Supplementary Figure 5. We did not mask the spatial domain used in the analysis based on cover fraction and threshold on the maximum percentage of missing data (compare all the

other maps included in main text and Supplementary Information that are masked on the correct spatial domain).

This oversight was only on the code for producing Supplementary Figure 5; all the numerical analyses on which the main text and figures are performed on the appropriate spatial domain that does not include these northernmost latitudes.

The correct visualization of the figure, masked according to the common spatial domain is shown below and appears consistent with the forest management map suggested by the reviewer, though with more conservative estimates of intact forests in the boreal zone: forest pixels with more than 50% missing data in the original data are excluded in our analyses (Methods).

Therefore, we confirm the correctness of our analyses and the robustness of our results and interpretation on intact and managed forests.

We stress that our approach to derive global distributions of managed and intact forests rely on a recent article focusing on the detection of global intact forests based on 30-meter satellite imagery²⁰. **The approach we adopted to retrieve managed and intact forest domains is consistent, transparent and reproducible** (see Methods).

Action taken. We have masked the intact/managed forest map on the correct spatial domain in the revised version (Supplementary Figure 10, updated number in the revised version) and briefly discussed in the method section the comparison with the map suggested by the reviewer (see lines 435-438).

Moreover, evidence from 30m resolution time series of forest recovery in boreal forests shows that recovery following harvest (i.e., management) is more rapid than recovery following wildfire (White et al. 2017 <http://dx.doi.org/10.1016/j.rse.2017.03.035>). How do

the authors reconcile this observation with their claim that resilience in managed boreal forests is lower than in intact forests?

It is difficult to compare the results of our study with those reported in the mentioned paper because **they refer to different processes** (recovery from harvest and fires versus recovery from continuous small climate perturbations) **and different definitions of forest resilience**. As in all previous comments, the reviewer considers resilience as the capacity to recover after a drastic shift in the forest state originating from a disturbance, while we (accordingly to the literature on the topic) define resilience as the capacity to recover from continuous small external perturbations mostly due to interannual variation in climate. See previous answers on this issue.

In addition, concerning the specific example proposed by the reviewer, we think that it is incorrect to derive the effect of management from a system exposed to different disturbances as they are proposing. The correct comparison would be for the same disturbance (e.g. fires) occurring on contrasting land management (intact versus managed forests). This is because different disturbances affect environmental conditions (e.g. soil, seed bank, etc) in a different way.

Page 5: “Results show that, at the global level, intact forests have a probability of AD conditional on high δTAC greater than 0.5 (Fig. 4a). This signal is statistically significant and increases with the severity of AD suggesting that the progressive deterioration of ecosystem states, as tracked by the decline of resilience, has likely contributed to the upsurge of negative anomalies. The emerging relation is mainly driven by boreal forests, particularly those in central Russia, where there is emerging a localized decline in forest resilience (Fig. 1c).” The drivers of the most abrupt declines (AD) in boreal forests of Canada and Russia (together by far the largest proportion of global boreal forests) will be wildfires and harvest. Since harvest always targets mature forests are you suggesting that mature forests that are approaching harvest age show signs of declining resilience? But, in your analyses you are not differentiating the drivers of AD.

In the paragraph just above that one mentioned by the reviewer we wrote: **“To exclude the effect of land management (e.g. apparent abrupt declines driven by forest harvest), we limited the analysis to global intact forests, with a focus on tropical and boreal regions that together cover about 97% of the investigated domain”**.

As clearly mentioned in the text, **the causal relation between changes in resilience and abrupt decline in the forest state has been conducted only on intact forests, where events of abrupt decline cannot be due to harvest. In managed forest areas considerations on the interplay between resilience, age and management are therefore not appropriate.**

Wildfire, in contrast to harvest, can affect stands of all ages. In boreal forests most of the area burned is ignited by lightning strikes whose location is not influenced by the “resilience” of forests. You may be observing some statistical correlations here, but I am not at all convinced that your signal has any ecological meaning or is an indicator of decline of forest resilience.

In this specific analysis, **we do not aim to attribute the causes of an abrupt decline in boreal forests, but to quantify if a sign of declining resilience can be statistically associated following abrupt shift in the system, regardless of the disturbance occurrence or type.** According to our results in boreal forests, this signal is significant ($p < 0.05$) and occurs with a probability of occurrence of about 0.65 (see Figure 3, Severity of AD of 6σ). This may occur for instance after prolonged plant water stress conditions which may reduce forest resilience and make them more vulnerable to a range of disturbances, including insect outbreaks (also mentioned as an additional important disturbance in boreal forests in one of the next comment by Rev. 3), droughts and wildfires²¹. Of course, there is a fraction of abrupt declines occurring without being preceded by a decline in forest resilience (e.g., wildfires ignited by lightning strikes as suggested by the reviewer); however, based on our results, these cases constitute a minor fraction for boreal forests. These later considerations have been discussed in the main text by focusing more specifically on tropical forests where these effects appear more important compared to boreal forests: “[...] *fast and strong disturbance events, such as fires²² or droughts²³, may induce an abrupt decline independently on long-term increasing trends in critical slowing down^{8,9}*”.

More specifically on the suitability of CSD-based resilience indicators (implemented in our study) to predict fire-induced tree mortality, we point out that a recent article has documented their potential¹¹. In the above-mentioned study, the authors found that the increased tree mortality risk observed in California, largely attributable to fire events, clearly showed a dependence on impaired hydraulic connectivity and reduced stomatal conductance, which are reflected in a loss of resilience over the period preceding the disturbance occurrence. Despite these findings not being directly extrapolated to the global scale, they do support the use of CSD-based resilience indicators to explore possible early warning signals of abrupt declines that may emerge from similar mechanisms in response to slow changing climate drivers.

Action taken. We have clarified the scope of the analysis (exploration of the causal link between decreasing resilience and abrupt decline) and improved the discussion in the main text (see lines 184-186 and lines 187-203).

On page 6 the authors write that: “These hypotheses are also consistent with the dominant climate drivers of δTAC in boreal and tropical regions (background climate and climate variability, respectively, Fig. 2b,c) and further supported by multiple independent evidences (e.g., refs(30,43–45)). Do the authors have any independent references for boreal forests? References 30, 43-45 are all studies from the tropics.

Action taken. We have added references (^{24,25}) in the revised version.

Page 6: “On the contrary, benefits induced by climate warming and CO₂ fertilization have outweighed such negative effects in boreal regions, ultimately leading to an improvement of forest resilience.” How do the authors reconcile large increases in the areas affected by wildfires and insects in global boreal forests with their claim that forest resilience has increase over the observation period? This contradiction has to be addressed by the authors.

This is not a contradiction. Consistent with the definition widely adopted in the literature (e.g., refs. (³⁻¹⁴)), forest resilience is not necessarily related to disturbance occurrences. Forest resilience indeed reflects the capacity of the forest ecosystems to recover from continuous

perturbations. Therefore, a decline in resilience may increase the susceptibility of forests to natural disturbances, but not all natural disturbances are associated with a loss in resilience. Indeed, many disturbances represent fast and abrupt shocks that are not anticipated by a change in resilience, so they may occur regardless on the changes in resilience. See our previous answers on this issue and ref. (9) for additional details on what regime shifts are announced by a decline in resilience. Therefore, **our observation-based results of declining forest resilience should not be interpreted as a direct indicator of increasing forest disturbances but instead they should be viewed as emergent signals of an increasing instability of forests.**

That said, even if at regional scales we observed an increase in resilience in boreal forests, we stress that there is a considerable fraction of high-latitude forests, prominently located in central Russia and western Canada (Fig. 1a), experiencing an opposite trend. We found that in these regions, the increasing instability was statistically associated with an abrupt decline in the forest state. This may indicate that in these zones the abrupt decline is following the drifting toward a critical threshold, which is likely triggered by the rapid changes in environmental drivers. Insect outbreaks promoted by conditions of water stress may plausibly represent one of the major disturbance agents, favoured by antecedent detrimental climatic conditions²⁶, which have ultimately caused such regime shifts. Such hypothesis are corroborated by independent observational evidences (refs. (27,28)).

These results emphasize the potential of CSD-based indicators as early warning signals of critical transitions in boreal forests particularly in view of the expected intensification of climate-driven disturbances owning the projected decline in water availability^{25,29}.

Action taken. These aspects have been clarified in the text (see lines 192-197). Furthermore, we have added a dedicated section in supplementary material to clarify this issue (Supplementary Methods 3 Forest resilience and disturbance regimes).

Concluding paragraph: “Therefore, it is becoming urgent to account for these trends in the design of effective forest-based mitigation strategies to avoid future negative surprises triggered by the increasing vulnerability of carbon stocks.” I completely agree with the statement. Forest mitigation strategies that do not consider vulnerability of carbon stocks, or worse, that lead to increased vulnerability of C stocks are counterproductive to effective climate mitigation. However, the authors have not yet convinced me that their measure of forest resilience is a suitable to detect the vulnerability of carbon stocks.

We appreciate that reviewer 3 shares our line of thought concerning the relevance of the vulnerability of the forest carbon stock under changing climate conditions. Understanding the complex pattern between climate variability and resilience of forests at the global scale is therefore an urgent but challenging task. As detailed in the above-mentioned answers to the reviewer’s concerns, we have used resilience indicators widely adopted in the literature (e.g., refs. (3-14)) and the best available Earth Observations to address urgent questions about emerging trends in ecosystem resilience. We believe that the clarification about the definition of resilience, the additional material provided in the main text and supplementary material and this response document have further clarified the methods, goals and assumptions of the study and provided additional evidences about its robustness and overall validity.

Three additional comments:

it is frustrating that authors submit a manuscript without line numbers. How are reviewers expected to comment on statements when all we can reference is a page number?

Action taken. We apologize for forgetting to include line numbers. We have added line numbers in the revised version of the manuscript.

There are a number of typos, grammatical errors and incorrect uses of words, which I would normally point out but without line numbers this is not worth my time.

Action taken. We have carefully checked the text and corrected the errors.

Figure 2b – the most common color-blindness in the human population is red-green. The authors could greatly improve this graph by replacing one of the red or green lines with a blue one and further modified the line types or added distinct markers so that readers do not have to rely on color alone. See also Nature 598, 224-225 (2021). doi: <https://doi.org/10.1038/d41586-021-02696-z>.

Action taken. According to the reviewer's suggestion, we have changed the line colors in Figure 1d (updated numbering of figures in the revised version).

References

1. Huete, A. *et al.* Overview of the radiometric and biophysical performance of the MODIS vegetation indices. *Remote Sens. Environ.* **83**, 195–213 (2002).
2. Camps-Valls, G. *et al.* A unified vegetation index for quantifying the terrestrial biosphere. *Sci. Adv.* **7**, eabc7447 (2021).
3. Scheffer, M. *et al.* Early-warning signals for critical transitions. *Nature* **461**, 53–59 (2009).
4. Verbesselt, J. *et al.* Remotely sensed resilience of tropical forests. *Nat. Clim. Change* **6**, 1028–1031 (2016).
5. Hirota, M., Holmgren, M., Nes, E. H. V. & Scheffer, M. Global Resilience of Tropical Forest and Savanna to Critical Transitions. *Science* **334**, 232–235 (2011).
6. Seddon, A. W. R., Macias-Fauria, M., Long, P. R., Benz, D. & Willis, K. J. Sensitivity of global terrestrial ecosystems to climate variability. *Nature* **531**, 229–232 (2016).
7. Scheffer, M., Carpenter, S., Foley, J. A., Folke, C. & Walker, B. Catastrophic shifts in ecosystems. *Nature* **413**, 591–596 (2001).
8. Scheffer, M., Carpenter, S. R., Dakos, V. & van Nes, E. H. Generic Indicators of Ecological Resilience: Inferring the Chance of a Critical Transition. *Annu. Rev. Ecol. Evol. Syst.* **46**, 145–167 (2015).
9. Dakos, V., Carpenter, S. R., van Nes, E. H. & Scheffer, M. Resilience indicators: prospects and limitations for early warnings of regime shifts. *Philos. Trans. R. Soc. B Biol. Sci.* **370**, 20130263 (2015).
10. Dakos, V. *et al.* Slowing down as an early warning signal for abrupt climate change. *Proc. Natl. Acad. Sci.* **105**, 14308–14312 (2008).
11. Liu, Y., Kumar, M., Katul, G. G. & Porporato, A. Reduced resilience as an early warning signal of forest mortality. *Nat. Clim. Change* **9**, 880–885 (2019).
12. Keersmaecker, W. D. *et al.* A model quantifying global vegetation resistance and resilience to short-term climate anomalies and their relationship with vegetation cover. *Glob. Ecol. Biogeogr.* **24**, 539–548 (2015).
13. Boers, N. Observation-based early-warning signals for a collapse of the Atlantic Meridional Overturning Circulation. *Nat. Clim. Change* **11**, 680–688 (2021).
14. Ciemer, C. *et al.* Higher resilience to climatic disturbances in tropical vegetation exposed to more variable rainfall. *Nat. Geosci.* **12**, 174–179 (2019).
15. Holling, C. S. Engineering Resilience versus Ecological Resilience. in *Engineering within Ecological Constraints* 31–43 (National Academy Press, 1996).
16. Scheffer, M. *et al.* Early-warning signals for critical transitions. *Nature* **461**, 53–59 (2009).

17. Trumbore, S., Brando, P. & Hartmann, H. Forest health and global change. *Science* **349**, 814–818 (2015).
18. Grimm, V. & Wissel, C. Babel, or the ecological stability discussions: an inventory and analysis of terminology and a guide for avoiding confusion. *Oecologia* **109**, 323–334 (1997).
19. Arnoldi, J.-F., Loreau, M. & Haegeman, B. Resilience, reactivity and variability: A mathematical comparison of ecological stability measures. *J. Theor. Biol.* **389**, 47–59 (2016).
20. Potapov, P. *et al.* The last frontiers of wilderness: Tracking loss of intact forest landscapes from 2000 to 2013. *Sci. Adv.* **3**, e1600821 (2017).
21. Forzieri, G. *et al.* Emergent vulnerability to climate-driven disturbances in European forests. *Nat. Commun.* **12**, 1081 (2021).
22. Brando, P. M. *et al.* Abrupt increases in Amazonian tree mortality due to drought–fire interactions. *Proc. Natl. Acad. Sci.* **111**, 6347–6352 (2014).
23. Doughty, C. E. *et al.* Drought impact on forest carbon dynamics and fluxes in Amazonia. *Nature* **519**, 78–82 (2015).
24. Scheffer, M., Hirota, M., Holmgren, M., Nes, E. H. V. & Chapin, F. S. Thresholds for boreal biome transitions. *Proc. Natl. Acad. Sci.* **109**, 21384–21389 (2012).
25. Gauthier, S., Bernier, P., Kuuluvainen, T., Shvidenko, A. Z. & Schepaschenko, D. G. Boreal forest health and global change. *Science* **349**, 819–822 (2015).
26. McDowell, N. G. *et al.* The interdependence of mechanisms underlying climate-driven vegetation mortality. *Trends Ecol. Evol.* **26**, 523–532 (2011).
27. Kurz, W. A. *et al.* Mountain pine beetle and forest carbon feedback to climate change. *Nature* **452**, 987–990 (2008).
28. Schaphoff, S., Reyer, C. P. O., Schepaschenko, D., Gerten, D. & Shvidenko, A. Tamm Review: Observed and projected climate change impacts on Russia’s forests and its carbon balance. *For. Ecol. Manag.* **361**, 432–444 (2016).
29. D’Orangeville, L. *et al.* Beneficial effects of climate warming on boreal tree growth may be transitory. *Nat. Commun.* **9**, 3213 (2018).

Reviewer Reports on the First Revision:

Referees' comments:

Referee #1 (Remarks to the Author):

I liked the original version of the manuscript and the revisions made have further improved it. This is a very interesting and timely manuscript and I now recommend publication in the present form.

Referee #2 (Remarks to the Author):

I want to thank the authors for taking the time and effort to address all my comments and suggestions.

Referee #3 (Remarks to the Author):

I thank the authors for their comprehensive responses to the suggestions of all three reviewers. I also am glad that the considerable efforts in reviewing this manuscript have helped identify a couple of errors and, to quote the authors' response, removed some "over interpretations of data". As a result of these revisions and the additional explanations and analyses provided in the Supplementary Information, the manuscript has improved considerably.

There remains some lack of clarity around the definition of the perturbations that are included in this analysis. This might be addressed, if some of the important statements in the newly added Methods 3 of the SI were moved to the main text of the manuscript.

The newly added section Methods 3 in the SI is very helpful. However, the essence of this information should really appear in the main text. In the analysis of forest resilience, this study does not deal with recovery after perturbations such as wildfire, insects or harvest but instead only addresses "short-term responses to weak continuous external perturbations". This is a somewhat arbitrary and unusual definition of resilience – and it should be made very clear that this is what the authors have chosen. The concluding statement of that methods section also needs to appear in the main text: "Therefore, declining forest resilience should not be interpreted as a direct indicator of increasing forest disturbances but instead they should be viewed as emergent signals of an increasing instability of forests."

Following up on the authors' choice of definition of resilience, what – if any – steps did the authors take to exclude in their analysis observations of the responses of forests to wildfire or major insect outbreaks? There is no explanation provided on how pixels with such disturbances were excluded. And if such pixels were not excluded, then the associated perturbations and subsequent recovery are also included in the analysis, and this should be made explicit along with an explanation of the consequences.

On line 453 the authors explain the methods for the assessment of the dTAC values prior to the observed abrupt declines. What would be the cause of the abrupt decline in intact forests since harvest does not occur in intact forests and the authors correctly state in Methods 3 that fires are not expected to be preceded by changes in dTAC values? How do the authors know whether the observed abrupt decline was triggered by a wildfire or some other process?

I can agree that there is more than one definition of resilience – but if the authors write that resilience is the capacity to withstand and recover from perturbations – then the authors should clearly explain which external perturbations they include and which external perturbations they exclude in their analysis.

Author Rebuttals to First Revision:

First of all we would like to thank the referees for their positive and constructive comments. In our revised version of the manuscript we addressed the additional suggestions raised by Rev. 3 in order to further improve the clarity of the text.

Referee #1

I liked the original version of the manuscript and the revisions made have further improved it. This is a very interesting and timely manuscript and I now recommend publication in the present form.

→ We thank the reviewer for the positive comment and the appreciation of our work.

Referee #2

I want to thank the authors for taking the time and effort to address all my comments and suggestions.

→ We thank the reviewer for the positive comment and the appreciation of our work.

Referee #3

I thank the authors for their comprehensive responses to the suggestions of all three reviewers. I also am glad that the considerable efforts in reviewing this manuscript have helped identify a couple of errors and, to quote the authors' response, removed some "over interpretations of data". As a result of these revisions and the additional explanations and analyses provided in the Supplementary Information, the manuscript has improved considerably.

→ We thank the reviewer for the positive comment.

There remains some lack of clarity around the definition of the perturbations that are included in this analysis. This might be addressed, if some of the important statements in the newly added Methods 3 of the SI were moved to the main text of the manuscript.

The newly added section Methods 3 in the SI is very helpful. However, the essence of this information should really appear in the main text. In the analysis of forest resilience, this study does not deal with recovery after perturbations such as wildfire, insects or harvest but instead only addresses "short-term responses to weak continuous external perturbations".

This is a somewhat arbitrary and unusual definition of resilience – and it should be made very clear that this is what the authors have chosen.

- ➔ **Action taken.** We have further clarified these concepts accordingly to the reviewer's suggestion (lines 59-61).
- ➔ We argue that the definition of resilience adopted in the work is not “*arbitrary and unusual*”, but consistent with a wide literature on the topic (e.g., refs. (1–12)). In our previous interaction with Rev. 3 we have provided a detailed response to this specific issue and described the additional sections in Supplementary Information about the scientific background (Supplementary Methods 1-3) on which our methodological framework is developed.

The concluding statement of that methods section also needs to appear in the main text: “Therefore, declining forest resilience should not be interpreted as a direct indicator of increasing forest disturbances but instead they should be viewed as emergent signals of an increasing instability of forests.”

- ➔ The content mentioned by the reviewer is already reported in the main text in lines 228-231, thus **we believe it is inappropriate to duplicate it in the conclusion section**. Hereafter the text already reported in the main text:

“We point out that these critical conditions are not sufficient to determine a regime shift (Supplementary Methods 3). However, they represent a strong indication of the rising risks of an increased instability and vulnerability to hazards of forest biomes.”

Following up on the authors' choice of definition of resilience, what – if any – steps did the authors take to exclude in their analysis observations of the responses of forests to wildfire or major insect outbreaks? There is no explanation provided on how pixels with such disturbances were excluded. And if such pixels were not excluded, then the associated perturbations and subsequent recovery are also included in the analysis, and this should be made explicit along with an explanation of the consequences.

- ➔ We thank the reviewer for rising this pertinent and relevant question, since state shifts may indeed affect temporal patterns in autocorrelation. Concerning the methodology, we decided not to exclude from the analysis sites where an explicit major disturbance occurred (e.g., insect outbreak, wildfire). Instead, we treated the time series of each site as a record of fluctuations of NDVI to any type of perturbation (strong or weak) and we measured the change of autocorrelation along the whole record. We adopted this methodology for the following reasons:
 - i) areas with major disturbances have low frequency and therefore do not likely affect global statistics;

- ii) large disturbances leading to abrupt declines (AD) in the vegetation state and following recoveries, as those potentially originating from forest disturbances, are distributed throughout the whole investigated period. Therefore, due to compensatory effects (i.e. positive trend in resilience of those affected at the beginning of the temporal window and negative trend of those affected at the end of the period), we expect that their overall effect on the average trend in autocorrelation is marginal.
- ➔ Following the request of the reviewer, we further assess with novel analyses and figures the robustness of our methodology and results with respect to this specific issue. To this scope, we tested two contrasting methods: 1) all areas affected by abrupt declines are included in the analyses (as reported in results shown in the main text), 2) all areas affected by abrupt declines are excluded from the analyses. As expected, the inclusion or exclusion of forest areas affected by abrupt declines does not affect the frequency distribution of the changes in temporal autocorrelation at the global scale and at climate region level (Extended Data Fig. 4k-o). Overall, we estimate a minor amplification (~1%) of the increase in temporal autocorrelation (reduction in resilience) when areas affected by AD are excluded from the analysis. This is further corroborated by the high consistency in the patterns of the trend in temporal autocorrelation in the climate space amongst the two AD experiments (Extended Data Fig. 5e-f).
- ➔ **These additional experiments further corroborate the robustness of our methodology and the validity of our results.**
- ➔ **Action taken.** We have described the new experiments and results in the Method section (lines 595-599, lines 605-606) and added a dedicated paragraph in Supplementary Discussion 2 “Sensitivity to the inclusion/exclusion of areas affected by abrupt declines”. New results are shown in Extended Data Fig. 4k-o and Extended Data Fig. 5e-f.

On line 453 the authors explain the methods for the assessment of the δTAC values prior to the observed abrupt declines. What would be the cause of the abrupt decline in intact forests since harvest does not occur in intact forests and the authors correctly state in Methods 3 that fires are not expected to be preceded by changes in δTAC values? How do the authors know whether the observed abrupt decline was triggered by a wildfire or some other process?

- ➔ The attribution of the cause of abrupt declines is out of the scope of this work, as already answered in our previous interaction with Rev. 3. As discussed in the main text we envisage that mortality due to climate drivers (e.g. water limitation) and or biotic agents (e.g. insect outbreaks) may lead to this type of events.
- ➔ Abrupt declines statistically associated to antecedent high values of δTAC are interpreted, consistently to the literature on the topic, as a drift of the ecosystem towards a critical resilience threshold plausibly associated to changes in

environmental drivers. These aspects have been already discussed in the main text (lines 187-203). For instance, when we interpreted results in boreal zones:

“Such patterns may indicate that in these zones the abrupt decline is following the drifting toward a critical threshold, which is likely triggered by the changes in environmental drivers occurring at the northernmost latitudes¹³. Insect outbreaks, which are typically favoured by forest water stress¹⁴, may represent one of the major disturbances that have ultimately caused such abrupt declines in the ecosystem state^{15,16}.”

→ **Action taken.** We have further clarify this issue in the method section (lines 680-682).

I can agree that there is more than one definition of resilience – but if the authors write that resilience is the capacity to withstand and recover from perturbations – then the authors should clearly explain which external perturbations they include and which external perturbations they exclude in their analysis.

- In accordance with the cited literature, resilience in the analysis is defined as the capacity of ecosystems to withstand from perturbations and avoid state shifts. Therefore, any type and magnitude of perturbation is accounted in the quantification of the temporal autocorrelation signal, in line with the literature on the subject.
- We believe that the additional clarifications provided along this revised version make this clearer.

References

1. Scheffer, M. *et al.* Early-warning signals for critical transitions. *Nature* **461**, 53–59 (2009).
2. Verbesselt, J. *et al.* Remotely sensed resilience of tropical forests. *Nat. Clim. Change* **6**, 1028–1031 (2016).
3. Hirota, M., Holmgren, M., Nes, E. H. V. & Scheffer, M. Global Resilience of Tropical Forest and Savanna to Critical Transitions. *Science* **334**, 232–235 (2011).
4. Seddon, A. W. R., Macias-Fauria, M., Long, P. R., Benz, D. & Willis, K. J. Sensitivity of global terrestrial ecosystems to climate variability. *Nature* **531**, 229–232 (2016).
5. Scheffer, M., Carpenter, S., Foley, J. A., Folke, C. & Walker, B. Catastrophic shifts in ecosystems. *Nature* **413**, 591–596 (2001).
6. Scheffer, M., Carpenter, S. R., Dakos, V. & van Nes, E. H. Generic Indicators of Ecological Resilience: Inferring the Chance of a Critical Transition. *Annu. Rev. Ecol. Evol. Syst.* **46**, 145–167 (2015).

7. Dakos, V., Carpenter, S. R., van Nes, E. H. & Scheffer, M. Resilience indicators: prospects and limitations for early warnings of regime shifts. *Philos. Trans. R. Soc. B Biol. Sci.* **370**, 20130263 (2015).
8. Dakos, V. *et al.* Slowing down as an early warning signal for abrupt climate change. *Proc. Natl. Acad. Sci.* **105**, 14308–14312 (2008).
9. Liu, Y., Kumar, M., Katul, G. G. & Porporato, A. Reduced resilience as an early warning signal of forest mortality. *Nat. Clim. Change* **9**, 880–885 (2019).
10. Keersmaecker, W. D. *et al.* A model quantifying global vegetation resistance and resilience to short-term climate anomalies and their relationship with vegetation cover. *Glob. Ecol. Biogeogr.* **24**, 539–548 (2015).
11. Boers, N. Observation-based early-warning signals for a collapse of the Atlantic Meridional Overturning Circulation. *Nat. Clim. Change* **11**, 680–688 (2021).
12. Ciemer, C. *et al.* Higher resilience to climatic disturbances in tropical vegetation exposed to more variable rainfall. *Nat. Geosci.* **12**, 174–179 (2019).
13. Serreze, M. C., Barrett, A. P., Stroeve, J. C., Kindig, D. N. & Holland, M. M. The emergence of surface-based Arctic amplification. *The Cryosphere* **3**, 11–19 (2009).
14. McDowell, N. G. *et al.* The interdependence of mechanisms underlying climate-driven vegetation mortality. *Trends Ecol. Evol.* **26**, 523–532 (2011).
15. Kurz, W. A. *et al.* Mountain pine beetle and forest carbon feedback to climate change. *Nature* **452**, 987–990 (2008).
16. Schaphoff, S., Reyer, C. P. O., Schepaschenko, D., Gerten, D. & Shvidenko, A. Tamm Review: Observed and projected climate change impacts on Russia’s forests and its carbon balance. *For. Ecol. Manag.* **361**, 432–444 (2016).